# Redistribution of EZH2 promotes malignant phenotypes by rewiring developmental programmes

Thomas Mortimer[1], Elanor N Wainwright[1], Harshil Patel[2], Bernard M Siow[3], Zane Jaunmuktane[4,5], Sebastian Brandner[5,6] & Paola Scaffidi[1,7,*]

## Abstract

**Epigenetic regulators are often hijacked by cancer cells to sustain malignant phenotypes. How cells repurpose key regulators of cell identity as tumour-promoting factors is unclear. The antithetic role of the Polycomb component EZH2 in normal brain and glioma provides a paradigm to dissect how wild-type chromatin modifiers gain a pathological function in cancer. Here, we show that oncogenic signalling induces redistribution of EZH2 across the genome, and through misregulation of homeotic genes corrupts the identity of neural cells. Characterisation of EZH2 targets in *de novo* transformed cells, combined with analysis of glioma patient datasets and cell lines, reveals that acquisition of tumorigenic potential is accompanied by a transcriptional switch involving de-repression of spinal cord-specifying *HOX* genes and concomitant silencing of the *empty spiracles* homologue *EMX2*, a critical regulator of neurogenesis in the forebrain. Maintenance of tumorigenic potential by glioblastoma cells requires *EMX2* repression, since forced *EMX2* expression prevents tumour formation. Thus, by redistributing EZH2 across the genome, cancer cells subvert developmental transcriptional programmes that specify normal cell identity and remove physiological breaks that restrain cell proliferation.**

**Keywords**  cancer; chromatin; EZH2; glioblastoma; Polycomb
**Subject Categories**  Cancer; Chromatin, Transcription & Genomics

## Introduction

Establishment and maintenance of cell identity within tissues is critical for proper organismal function. Regulatory mechanisms that specify cell fate during embryogenesis involve a complex interplay between morphogens, transcription factors and epigenetic regulators that modulate chromatin structure and DNA methylation patterns [1]. The coordinated action of these three layers of regulation establishes cell type-specific transcriptional programmes and specifies cell fate [1]. Similar mechanisms persist into adulthood, and control adult stem cell function and tissue homeostasis [2]. Both in the embryo and in the adult, lineage commitment entails three major steps: (i) restriction of cellular plasticity, achieved through silencing of pluripotent/multipotent genes and repression of alternative lineage-specific genes, (ii) activation of lineage-specific transcriptional programmes mediated by key transcription factors, and (iii) a transition from proliferation to differentiation via regulation of self-renewal transcriptional programmes [3]. A key role in this process is played by chromatin structure and DNA modifications, which together modulate accessibility of transcription factors to gene regulatory elements [4].

Whilst the role of epigenetic regulators in development and stem cell regulation has long been appreciated, a surprising "double-life" for these proteins in cancer has recently emerged. Numerous epigenetic regulators have been shown to have a tumour-promoting role in various malignancies and be critically required for tumour maintenance [5]. Examples of such proteins include chromatin modifiers, chromatin remodellers and histone modification "readers" [5]. In all cases, inhibition of protein function, through either genetic or pharmacological means, severely impairs disease maintenance, indicating a dependency of cancer cells on these factors for survival and proliferation [5]. Remarkably, epigenetic regulators often exert such a tumour-promoting role in their wild-type state, in the absence of any mutation affecting their intrinsic function [5]. Thus, proteins that play key physiological roles in defining normal cell identity acquire a different, pathological function in transformed cells.

A striking example that illustrates the dichotomous role of epigenetic regulators in physiology and cancer is enhancer of zeste homologue 2 (EZH2), particularly when examining its function within the central nervous system (CNS). EZH2 is the catalytic component of the Polycomb repressive complex 2 (PRC2) and is responsible for

1   Cancer Epigenetics Laboratory, The Francis Crick Institute, London, UK
2   Bioinformatics and Biostatistics, The Francis Crick Institute, London, UK
3   In Vivo Imaging, The Francis Crick Institute, London, UK
4   Department of Clinical and Movement Neurosciences, Queen Square Brain Bank, UCL Queen Square Institute of Neurology, London, UK
5   Division of Neuropathology, National Hospital for Neurology and Neurosurgery, University College London Hospitals NHS Foundation Trust, London, UK
6   Department of Neurodegenerative Disease, UCL Queen Square Institute of Neurology, London, UK
7   UCL Cancer Institute, University College London, London, UK
    *Corresponding author. Tel: +44 020 3796 1325; E-mail: Paola.Scaffidi@crick.ac.uk

the deposition of a tri-methyl mark at lysine 27 of histone H3 (H3K27me3) [6]. H3K27me3 mediates gene repression through a complex interplay between PRC2 and the Polycomb repressive complex 1 (PRC1). Evidence from various systems suggests a hierarchical model of Polycomb function, whereby PRC2-deposited H3K27me3 recruits PRC1, which in turn induces chromatin compaction and inhibits the activity of RNA polymerase II [7]. However, reciprocal recruitment of PRC2 via the PRC1-deposited H2AK119ub has recently been documented, suggesting that regulation of gene expression programmes requires a complex series of interactions between PRC1 and PRC2 [8–10]. EZH2 is widely expressed in the CNS during development and is essential for correct specification of regional and cellular identity [11–13]. In the forebrain, EZH2 regulates both self-renewal of neural stem cells (NSCs) and the balance between neurons and glial cells, primarily by repressing lineage-specific transcription factors until the correct stage of development is reached [11,14]. Through a similar mechanism, EZH2 contributes to motor neuron subtype specification in the spinal cord, where it represses *HOX* genes in a region-specific manner and maintains sharp expression domains for this critical group of transcription factors [12]. Thus, EZH2's primary function in the developing CNS is to prevent inappropriate expression of developmental regulators and ensure that cell type-specific transcriptional programmes are executed at the correct stage of development and in the correct CNS region.

EZH2 also supports brain function in the adult. After birth, EZH2 is highly expressed in cells located in the subventricular zone (SVZ), where it continues to regulate neurogenesis [15,16]. In addition to its role in regulating CNS development and maintenance, recent evidence suggests that EZH2 also exerts an important tumour-suppressive function in the brain. Dominant-negative inhibition of PRC2 activity by recurrent H3K27 mutations drives the development of paediatric glioma [17], and EZH2-deficient mice show accelerated and more aggressive development of myc-driven medulloblastoma [18]. Furthermore, damaging mutations affecting EZH2 and other PRC2 components are recurrently observed in glioblastoma multiforme (GBM; WHO grade IV) patients, suggesting that normal cells use EZH2 to counteract oncogenic challenges [19,20]. However, strong evidence suggests that EZH2 acquires a distinct, tumour-promoting role in malignant neural cells, as inhibition of its function impairs the maintenance of various CNS cancers [21–23]. EZH2 appears to be particularly important in high-grade gliomas where Polycomb repressive complexes promote disease progression and therapy resistance by sustaining cancer cell self-renewal and favouring cellular plasticity [24–29]. These observations suggest that cancer cells which retain a functional PRC2 hijack EZH2 and corrupt its function to promote tumour maintenance. Notably, the dichotomous role of EZH2 in physiology and cancer is not restricted to the nervous system and is observed in several other tissues, suggesting that common principles may underlie the switch to a pathological function in various cellular contexts [30].

In this study, using EZH2 as a paradigm, we set out to understand how epigenetic regulators that play essential roles in establishing and maintaining normal cell identity are repurposed by cancer cells as tumour-promoting factors. We find that redistribution of EZH2 across the genome in transformed cells induces misregulation of surprisingly few, but key, regulators of neural developmental programmes, resulting in aberrant cell identity and unrestricted

proliferation. Thus, by redistributing EZH2 on chromatin, cancer cells remove physiological breaks that normally restrain cellular plasticity and enhance their malignant phenotypes. Since maintenance of these rewired transcriptional programmes is required for tumour growth, cells become dependent on EZH2 and thus vulnerable to its inhibition.

# Results

## Neoplastic transformation changes EZH2 chromatin binding profiles

Characterisation of the mechanisms underpinning the hijacking of EZH2 in human neural cancers requires direct comparison of normal and malignant cells. A challenge in doing so is that the identity of the cell responsible for initiating the disease is unclear. For example, medulloblastoma may arise from multiple cell populations, located either within the cerebellum or in the dorsal brainstem [31]. Similarly, the cellular origin of gliomas remains a topic of controversy and the high degree of molecular and clinical heterogeneity observed in patients is thought to reflect the diverse cell types that can initiate the disease [32]. This uncertainty regarding the cancer cell-of-origin hinders accurate modelling of neural neoplastic transformation. Furthermore, isolation of normal neural cells of human origin from adult individuals presents major challenges, precluding direct comparison of normal and cancerous cells. We therefore opted to begin our investigation using a well-characterised and isogenic model of cancer development previously shown to be relevant for glioma [33], in which fibroblastic cells are *de novo* transformed by inactivation of p53 and pRB tumour suppressors and activation of RAS signalling [34], events which recurrently occur in GBM [35,36](Fig 1A). Although atypical as a choice to study brain-related processes, this experimental system has proven useful to discover GBM-relevant mechanisms, as *de novo* transformed fibroblasts acquire several phenotypic and functional features that characterise glioma stem cells [33,37–39]. Furthermore, a major subtype of GBM is characterised by mesenchymal features and expression of several fibroblastic markers [36,40], and regardless of the molecular subtype, mesenchymal traits are associated with resistance to therapy in GBM patients, indicating their clinical relevance [41,42]. Based on this knowledge, we decided to take advantage of the fibroblast-based system's tractability to reveal candidate EZH2-related mechanisms, and subsequently validate them in GBM cells.

To examine how EZH2 function is affected by neoplastic transformation, we characterised three cellular states generated by sequential modification of primary fibroblasts: untransformed cells, immortalised by expression of human telomerase (hTERT) to avoid confounding effects associated with replicative senescence of primary cell populations; pre-neoplastic cells, with inactivated p53 and pRb, but lacking tumorigenic potential; and transformed cells, which also express oncogenic HRAS$^{v12}$ and induce tumour formation when injected into immunocompromised mice (Fig 1A) [34]. Quantification of EZH2 and H3K27me3 levels showed a progressive increase in EZH2 levels in pre-neoplastic and transformed cells but unaltered levels of H3K27me3, in line with the notion that highly

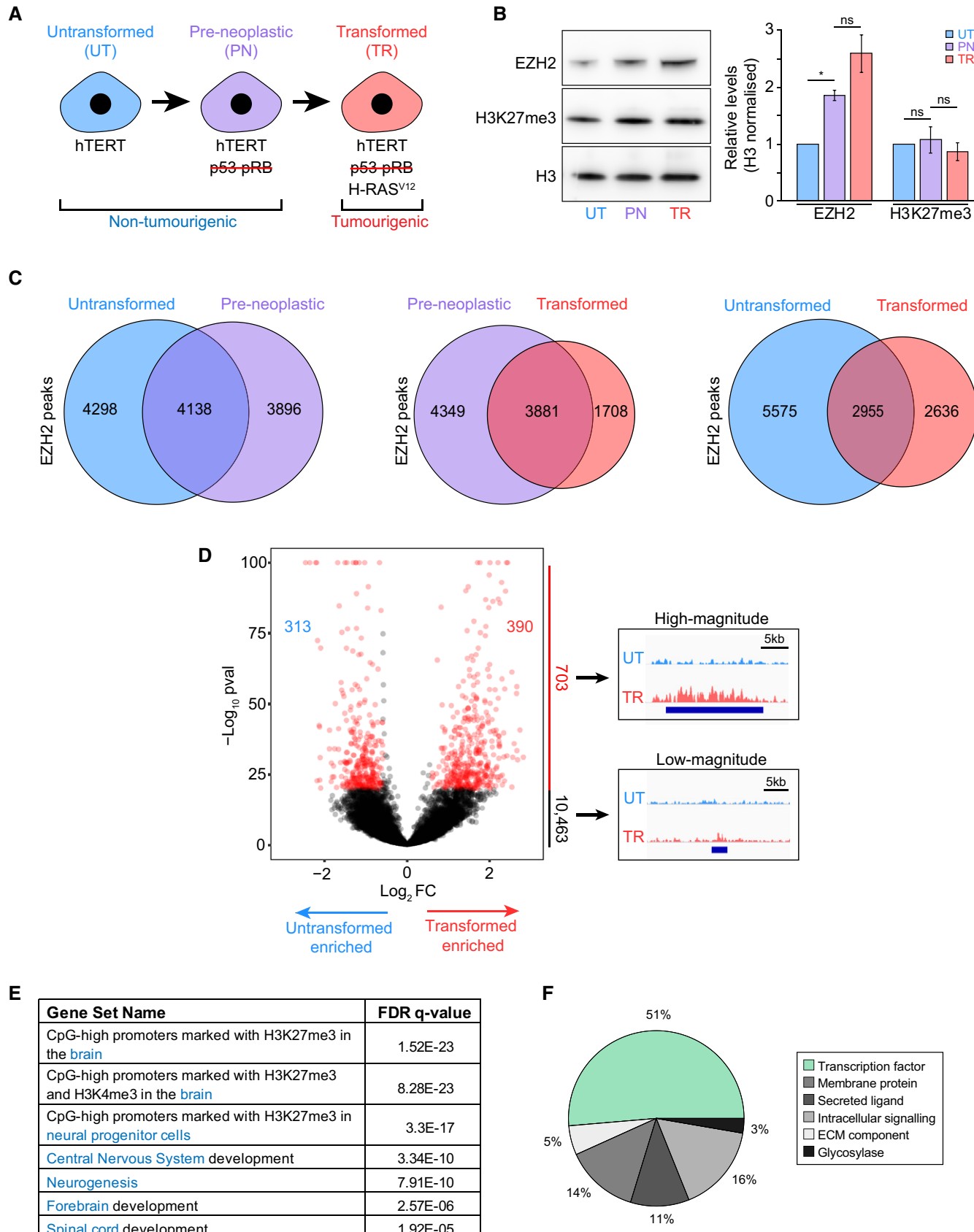

Figure 1.

**Figure 1.** ***De novo* transformation drives extensive redistribution of EZH2.**

A  Schematic representation of the fibroblast-based model of cancer development used in this study. Red strikethrough represents inhibition of p53 and pRB by SV40 large and small T-antigens.

B  Quantification by Western blot analysis of EZH2 and H3K27me3 levels at each stage of *de novo* transformation. UT, untransformed; PN, pre-neoplastic; TR, transformed. Histone H3 is used as a loading control. The graph on the right displays the H3 normalised densitometric values of Western blot bands from untreated and DMSO-treated cells. Values represent mean ± SEM from three biological replicates. One asterisk indicates *P*-value < 0.05 (one-tailed unpaired Student's *t*-test). ns, non-significant.

C  Venn diagrams showing the overlap between H3K27me3-associated EZH2 binding sites detected at each stage of *de novo* transformation.

D  Volcano plot showing the relative enrichment of EZH2 binding at all sites detected in either untransformed or transformed cells. Numbers within the volcano plot indicate differential binding sites with a *P*-value ≤ 1e-20 and a fold change in normalised tag count ≥ 1.5. Enrichment significance calculated based on a Poisson distribution using the "getDifferentialPeaks" function in HOMER (see Materials and Methods). *P*-values of > 1e-100 were set to 1e-100 for display reasons. Tracks on the right show representative high- and low-magnitude EZH2 differential peaks. ChIP-seq signal normalised to sequencing depth is shown. Tracks are scaled to be of the same height to make samples comparable. Blue bars represent regions called as an EZH2 binding site. UT, untransformed; TR, transformed.

E  Neural-related GSEA gene signatures enriched amongst genes with a high-magnitude, differential EZH2 binding site at their promoter [±5 kb transcription start site (TSS)]. Gene signatures are derived from GSEA Curated and Gene Ontology gene sets. The following gene signature names were modified from their original molecular signature database descriptor for clarity: "CpG high promoters marked with H3K27me3 in the brain" - "MEISSNER_BRAIN_HCP_WITH_H3K27ME3", "CpG high promoters marked with H3K27me3 and H3K4me3 in the brain" – "MEISSNER_BRAIN_HCP_WITH_H3K4ME3_AND_H3K27ME3", "CpG high promoters marked with H3K27me3 in neural progenitor cells" - "MIKKELSEN_NPC_HCP_WITH_H3K27ME3".

F  Functional classification of differential site-associated genes (total genes: 37) that are found in the gene signatures "Central Nervous System Development" and "Neurogenesis".

proliferative cells upregulate EZH2 to maintain homeostatic levels of H3K27me3 [43] (Fig 1B).

We then explored whether EZH2 distribution on chromatin was affected by transformation. To do so, we performed chromatin immunoprecipitation sequencing (ChIP-seq) and mapped EZH2 binding sites and its associated H3K27me3 mark across the three cellular states (Figs 1C and EV1A). As expected, in all conditions, the distribution of EZH2 and H3K27me3 was highly concordant, with more than 95% of EZH2 binding sites overlapping a H3K27me3 peak (Fig EV1B and C). The number of H3K27me3-associated EZH2 sites was comparable across cellular states, ranging from ~6,000 to 8,000. However, detected binding sites only showed a partial overlap amongst conditions, with a substantial fraction of binding sites appearing or disappearing at any transition (Fig 1C). In particular, the untransformed and transformed states shared less than 50% of EZH2 binding sites, indicating that oncogenic signalling induced extensive redistribution of EZH2 and its associated mark across the genome (Fig 1C). Thus, whilst EZH2 activity does not change upon transformation, its redistribution creates distinct domains of repressive chromatin in normal and neoplastic cells.

Although thousands of EZH2 binding sites were either lost or gained upon transformation, we reasoned that not all changes would likely have biological consequences. We therefore employed a multi-step filtering strategy to identify functionally important sites. In a first step, we searched for high-magnitude peaks, indicative of strong EZH2 binding, that changed substantially upon transformation. To do so, we calculated the relative fold change in EZH2 signal at binding sites detected in each of the three transformation states (Figs 1D and EV1D). Although many regions showed differential EZH2 signal between states, in agreement with the large number of uniquely detected binding sites, large changes in EZH2 binding (≥ 1.5-fold, *P*-value ≤ 1e-20) were only observed at a small minority of sites (1–6% depending on the transition; Figs 1D and EV1D). Overall, out of 11,166 binding sites detected in untransformed or transformed cells only 313 showed substantially decreased EZH2 binding upon transformation (untransformed-specific sites), whereas 390 displayed strongly enhanced binding (transformed-specific sites) (Fig 1D). Of these large-magnitude differential binding

sites, 87% were also identified in additional ChIP-seq replicates of EZH2, confirming robust detection of differential sites (Fig EV1E). Interestingly, in light of the known preference of EZH2 for GC-rich sites [44], transformed-specific EZH2 binding sites were depleted of CpG island-containing regions compared to common (*P* < 0.0001, Fisher's test) and untransformed-specific sites (*P* < 0.0001, Fisher's test; Fig EV1F). This observation suggests the presence of distinct targeting mechanisms in transformed cells, which may redistribute EZH2 in a GC-independent manner. Furthermore, a large proportion of differential EZH2 binding sites were located within lamina-associated domains [45], with untransformed-specific sites being significantly depleted (*P* < 0.0001, Fisher's test) and transformed-specific sites being enriched (*P* < 0.0001, Fisher's test) compared to common sites (Fig EV1G). This observation suggests that genomic regions located at the nuclear periphery are particularly affected by transformation-driven changes in H3K27me3 levels—an interesting pattern considering that changes in gene–lamina interactions reflect transitions in cell identity [46–48].

To begin to shed light on the functional consequences that transformation-driven redistribution of EZH2 may have on cellular states, we examined which genes associate with the identified differential binding sites (see Materials and Methods). Gene set enrichment analysis revealed a significant enrichment (FDR > 1e-5) of neural-related signatures, including signatures of central nervous system development and neurogenesis (Fig 1E). Of note, more than 50% of neural-related, differentially bound loci were transcription factor-encoding genes (Fig 1F). Together with confirming the relevance of the model system to study brain-related processes, these observations suggest that EZH2 redistribution may rewire core neural development programmes during neoplastic transformation.

**Extensive redistribution of EZH2 upon transformation affects the expression of surprisingly few, but important genes**

Having identified major EZH2 binding sites that are either gained or lost upon transformation, we applied a second filter and searched, amongst those, for changes associated with relevant alterations in gene expression. To do so, we performed RNA-seq analysis to identify

genes that fulfilled three different criteria: genes had to (i) respond to inhibition of EZH2 by the specific inhibitor EPZ-6438 (EZH2i) [21], (ii) have a differential EZH2 peak at their promoter (±5 kb from the TSS), as defined by the analysis described above, (iii) show transcriptional changes consistent with altered EZH2 binding upon transformation (i.e. downregulation when EZH2 binding sites were gained in transformed cells, and upregulation when sites were lost).

EZH2i treatment effectively inhibited EZH2 activity and induced a strong reduction in H3K27me3 levels in all cellular states, leading to differential expression of 900–1,200 genes in each condition (Figs 2A and B, and EV2A). More than 80% of differentially expressed genes were upregulated by EZH2i, including both direct and indirect EZH2 targets (Fig 2B). Notably, only 35% of genes were commonly upregulated in the untransformed and transformed states, showing that the set of EZH2-sensitive genes changed substantially upon transformation (Fig 2C). Surprisingly, only a minority of genes bound by EZH2 at their promoter (14% for untransformed and 23% for transformed cells) responded to EZH2i treatment, suggesting that removal of H3K27me3 was not sufficient to relieve gene repression (Fig EV2B). EZH2i-insensitive genes did not show distinct EZH2 or H3K27me3 patterns at their promoter (Fig EV2C), but were characterised by high levels of DNA methylation, as indicated by bisulphite sequencing analysis of selected genes (Fig EV2D). In agreement, six distinct GBM cell lines showed significantly higher levels of DNA methylation at EZH2i insensitive genes compared to genes that responded to the inhibitor (Fig EV2E), and a similar pattern was observed in 51 cell lines from seven other cancer types (Fig EV2F). These observations suggest that silencing of many EZH2-bound genes is achieved through redundant mechanisms, and only genes exclusively reliant on PRC2 are de-repressed upon loss of EZH2 binding or inhibition of its activity. Thus, although EZH2 undergoes extensive redistribution upon transformation, the presence of redundant repressive mechanisms limits the number of genes affected by these changes at the transcriptional level (21 and 26 in untransformed and transformed cells, respectively; Fig 2D).

As a final filter to narrow down critical targets that may explain the switch from physiological to pathological function for EZH2, we eliminated genes that did not show the expected transcriptional changes upon transformation (Table EV1). The final list contained 14 EZH2 target genes specific for untransformed cells and seven specific for transformed cells (Fig 2D and E, and Table EV1). This small set of EZH2 targets contained multiple key regulators of neurogenesis, including various transcription factors (*HOXA11*, *HOXB9*, *NR6A1*, *SIM2*, *EMX2*), the PRC1 component *CBX2*, and other proteins involved in neuronal function (*CACNG8*, *SLC30A3*, *TENM4*) or GBM development (*PREX1*, *BCL2*; Fig 2E). Interestingly, the final list of EZH2 differential targets did not include *CDKN2A*/p16, previously implicated as a key EZH2 target gene in cancer cells [28,49,50]. *CDKN2A*/p16 was not bound by EZH2 in any cellular state, and its expression did not substantially change across transformation or upon EZH2i treatment, indicating that acquisition and maintenance of tumorigenic potential do not require PRC2-mediated repression of *CDKN2A*/p16 (Fig EV2G).

## EZH2 redistribution causes a transcriptional switch of homeotic genes

In light of their critical role in brain development, we focused our attention on two homeotic genes differentially regulated by EZH2 in untransformed and transformed cells: homeobox B9 (*HOXB9*) and empty spiracles homeobox 2 (*EMX2*; Fig 3A). HOXB9 and EMX2 play distinct, non-overlapping functions in specifying regional identity during CNS development. EMX2 plays a prominent role in the regulation of neurogenesis in the developing forebrain [51], whilst HOXB9 is involved in motor neuron subtype specification in the developing spinal cord [52] (Fig 3A). Both in the embryo and in the adult, EMX2 is expressed in NSCs where it restrains cell proliferation by regulating the balance between symmetric and asymmetric division [53–55]. Precise regional expression is essential for *HOX* gene function, and aberrant ectopic expression in the CNS leads to various abnormalities, including homeotic transformations and switches in cellular identity [56,57].

*EMX2* was highly expressed in untransformed fibroblasts but underwent a strong downregulation specifically in the transition from pre-neoplastic to tumorigenic transformed cells, consistent with the appearance of a large domain of EZH2-bound chromatin enriched for H3K27me3 (Fig 3B). Conversely, loss of H3K27me3 at *HOXB9* in transformed fibroblasts correlated with de-repression of the gene (Figs 3B and EV3A). RT-qPCR on cells treated with EZH2i confirmed the RNA-seq results showing upregulation of *HOXB9* in untransformed cells and re-expression of *EMX2* in transformed cells (Fig 3C), thereby validating the genes as bona fide EZH2 targets in the relevant cellular states. In further support, genetic inactivation of *EZH2* by CRISPR-mediated knock-out in transformed cells led to *EMX2* de-repression (Appendix Fig S1A). We conclude that altered binding of EZH2 to chromatin upon transformation leads to aberrant silencing of *EMX2* and concomitant de-repression of *HOXB9*.

Characterisation of normal brain cells and glioma cell lines confirmed the observations made using *de novo* transformed fibroblasts. Analysis of public mRNA expression data showed expression of *EMX2* in both embryonic and adult astroglia, which includes NSCs [15,58], whilst *HOXB9*, as expected, was repressed in all analysed cell types (Fig 3D). This expression pattern is consistent with observations made in the murine CNS [52,55]. Conversely, expression data from the cancer cell line encyclopaedia database (CCLE) showed widespread repression of *EMX2* in a large panel of glioma cell lines, whilst aberrant expression of *HOXB9* was detected in 19 lines (RPKM ≥ 5; Fig 3E). In line with a general decay of repressive chromatin at *HOX* clusters upon *de novo* transformation (Fig EV3A), we observed aberrant expression of most *HOX* genes across numerous glioma lines (Fig EV3B). Although genes within the *HOXB* cluster showed overall highest levels of expression, all clusters were affected, indicating general de-repression of *HOX* genes in glioma (Fig EV3B). As predicted, ChIP-PCR showed binding of EZH2 at the promoter of *EMX2* in M059K GBM cells, whilst EZH2 signal was minimal at *HOXB9* (Fig 3F). Furthermore, treatment of five distinct GBM lines with EZH2i induced expression of *EMX2*, confirming direct repression by EZH2 (Fig 3G). Interestingly, the magnitude of *EMX2* upregulation upon EZH2i treatment varied from a 2-fold to 250-fold change in the different GBM cell lines, and inversely correlated with the degree of DNA methylation at the *EMX2* promoter, again suggesting that DNA methylation could act as a redundant mechanism to repress PRC2 targets (Fig 3H). The observation that GBM cells employ multiple mechanisms to repress *EMX2* also suggests that preventing expression of *EMX2* may be particularly important to preserve their malignant phenotype. To examine the relevance of the EZH2-*EMX2* link more broadly in cancer, we

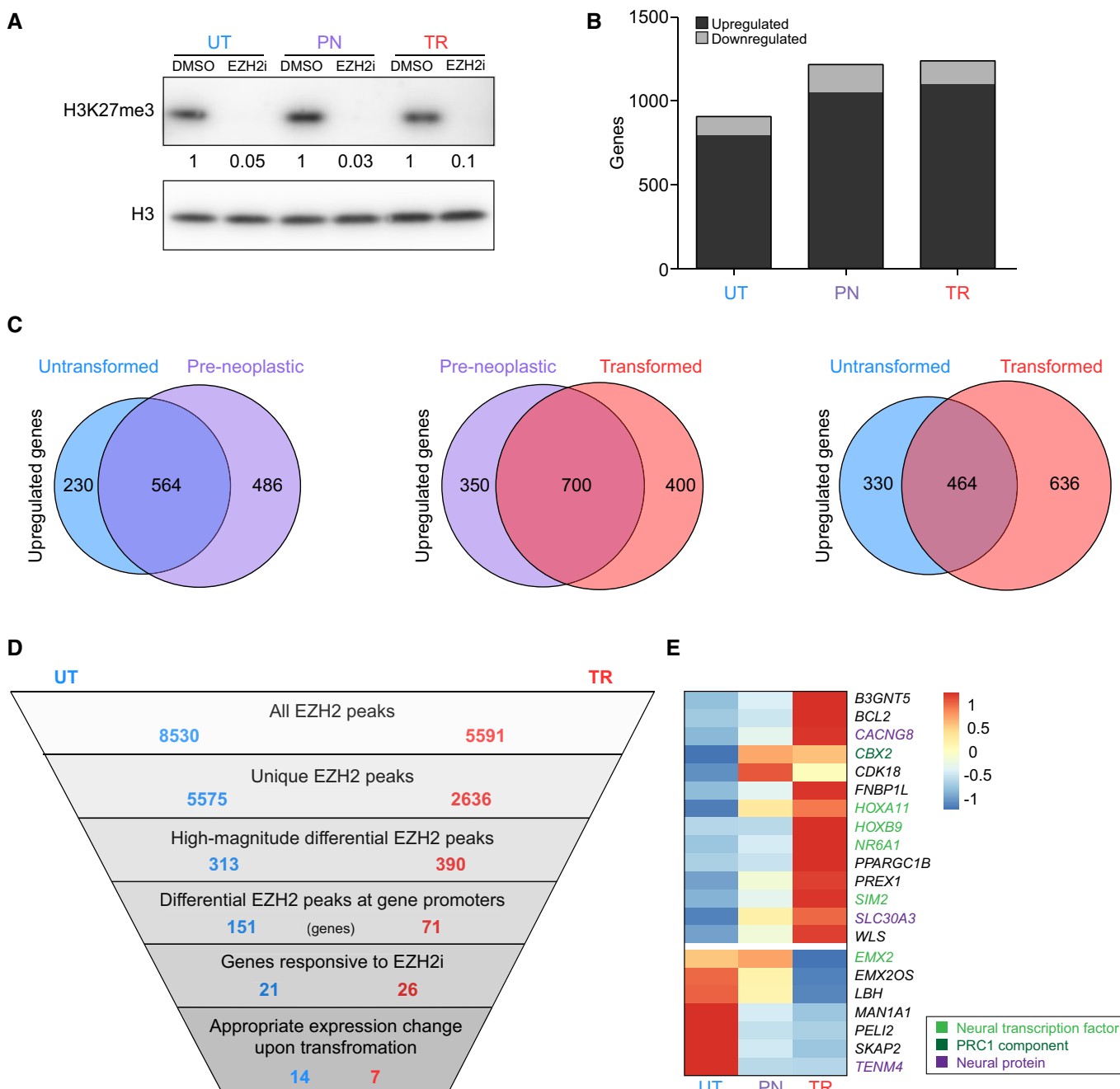

**Figure 2. Redistribution of EZH2 leads to misregulation of key developmental regulators.**

A   Western blot analysis of H3K27me3 levels after treatment of cells with an EZH2 inhibitor (EZH2i) or DMSO as control for 12 days. Histone H3 is used as a loading control. Values represent H3-normalised densitometric values of the H3K27me3 bands and are expressed relative to the relevant DMSO control. UT, untransformed; PN, pre-neoplastic; TR, transformed.

B   Quantification of differentially expressed genes [false discovery rate (FDR) ≤ 0.01], $Log_2FC \geq 1/\leq -1$ and maximal transcripts per million (maxTPM) ≥ 1) detected by RNA-seq in each cellular state upon EZH2i treatment. UT, untransformed; PN, pre-neoplastic; TR, transformed.

C   Venn diagrams showing the overlap between genes upregulated (FDR ≤ 0.01, $Log_2FC \geq 1$ and maxTPM ≥ 1) by EZH2i treatment in each cellular state.

D   Schematic representation of the multi-step filtering strategy employed to identify functionally important changes in EZH2 distribution induced by neoplastic transformation. Blue and red numbers represent the number of peaks/genes present after each filtering step in untransformed and transformed cells, respectively. UT, untransformed; TR, transformed.

E   Heatmap showing the relative expression of genes identified by the multi-step filtering strategy in the indicated cellular states. UT, untransformed; PN, pre-neoplastic; TR, transformed. Colours represent row-centered TPM values.

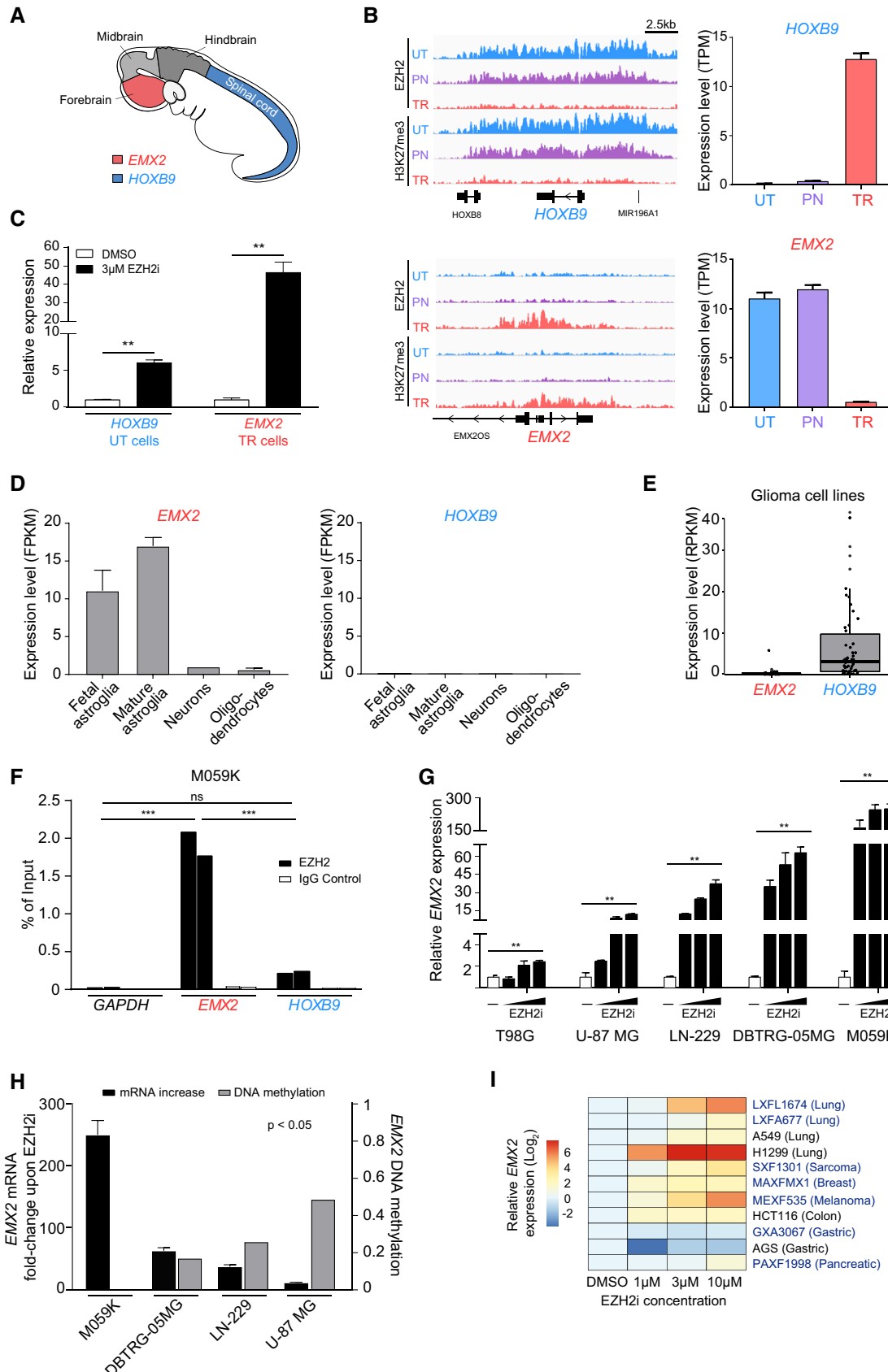

Figure 3.

**Figure 3.  EZH2 mediates an *EMX2*-to-*HOXB9* transcriptional switch upon neoplastic transformation.**

A   Schematic representation of *EMX2* and *HOXB9* expression in the embryonic nervous system.
B   ChIP-seq signal for EZH2 and H3K27me3 (left) at *EMX2* and *HOXB9* loci. ChIP-seq signal normalised to sequencing depth is shown. Tracks are scaled to be of the same height to make samples comparable. mRNA expression of *EMX2* and *HOXB9* (right) as detected by RNA-seq. The expression values represent mean ± SEM from three biological replicates. UT, untransformed; PN, pre-neoplastic; TR, transformed; TPM, transcripts per million.
C   RT–qPCR showing the expression levels of *EMX2* and *HOXB9* in the indicated cells treated with EZH2i or a DMSO control for 12 days. Values represent mean ± SEM from three biological replicates. Two asterisks indicate *P*-value < 0.01 (one-tailed unpaired Student's *t*-test). UT, untransformed; TR, transformed.
D   The expression levels of *EMX2* and *HOXB9* in primary human neural cells, as detected by RNA-seq. Data sourced from Brainseq2. The expression values represent mean ± SEM from four, twelve, one and five biological replicates of foetal astrocytes, mature astrocytes, neurons and oligodendrocytes, respectively. FPKM, fragments per kilobase of transcript per million.
E   The expression levels of *EMX2* and *HOXB9* in 62 glioma cell lines. Data sourced from CCLE. Every dot represents a cell line. RPKM, reads per kilobase of transcript per million. In the boxplot, the top, middle and bottom box delimiters represent the 75th, 50th and 25th percentiles of the data, respectively. Top and bottom whiskers show the 75th percentile + 1.5*interquartile range and 25th percentile − 1.5*interquartile range, respectively.
F   Quantification of EZH2 binding at *EMX2*, *HOXB9* and *GAPDH* (negative control) promoters by ChIP-qPCR in M059K GBM cells. Values from two biological replicates are shown. Three asterisks indicate *P*-value < 0.001 of EZH2 samples relative to the negative control GAPDH (two-way ANOVA followed by pairwise comparisons using Holm–Sidak method). Ns, non-significant.
G   Quantification of *EMX2* levels by RT–qPCR in five different GBM cell lines upon treatment with 1, 3 or 10 µM of EZH2i for 8 days, or a DMSO control. Values represent mean ± SEM from three technical replicates. The SEM is indicated to show reliability of the RT–qPCR values, due to the low endogenous levels of *EMX2*. Two asterisks indicate *P*-value < 0.01 comparing EZH2i- and DMSO-treated cells (two-way ANOVA).
H   Relationship between the extent of *EMX2* upregulation induced by EZH2i and the DNA methylation level at the *EMX2* promoter in GBM cell lines. Values for *EMX2* upregulation are the fold change in mRNA expression induced upon treatment with 10 µM EZH2i (see: panel G) and are expressed relative to a DMSO control. Values represent mean ± SEM from three technical replicates. The SEM is indicated to show reliability of the RT–qPCR values, due to the low endogenous levels of *EMX2*. Methylation values are averages of DNA methylation at CpGs across the promoter of *EMX2* in GBM cell lines. All data sourced from the CCLE. The significance of the anti-correlation between mRNA levels and DNA methylation levels across cell lines is indicated (Spearman rank correlation).
I   Heatmap visualising the fold change in *EMX2* expression as detected by RT–qPCR after treatment of the indicated PDX-derived (blue) and established cancer cell lines (black) with EZH2i for 8 days. Values are from three technical replicates.

treated a panel of cell lines isolated from patient-derived xenograft (PDX) models and additional established cancer cell lines with EZH2i (Fig 3I). EZH2i led to *EMX2* de-repression in nine cell lines from six cancer types, suggesting that cancer cells may generally use EZH2 to silence *EMX2*. Taken together, these results indicate that transformation-driven redistribution of EZH2 leads to aberrant regulation of key homeotic genes in neural cells, inducing silencing of a forebrain-specific transcription factor and ectopic expression of spinal cord-specific regulators.

**Misregulation of *EMX2* and *HOX* genes in glioma patients**

To determine the clinical relevance of the observed *EMX2-HOX* switch mediated by EZH2 redistribution, we examined expression patterns in publically available datasets from GBM patients. For this analysis, we primarily used the Repository of Molecular Brain Neoplasia Data (REMBRANDT) dataset [59], as it is the largest available RNA-seq dataset including normal controls, and subsequently confirmed our findings using additional datasets. In agreement with the results obtained with glioma cell lines, *EMX2* was significantly repressed in tumour samples compared to normal individuals (Fig 4A; *P* < 0.0001), whilst numerous *HOX* genes across all clusters showed higher levels in patients (Fig 4B).

*EZH2* and *EMX2* levels showed a significant anti-correlation in multiple glioma patient datasets, supporting the hypothesis that EZH2 represses *EMX2* in patients (Figs 4C, and EV4A and C). Information about GBM molecular subtypes available in the TCGA dataset allowed us to examine whether the *EZH2-EMX2* link correlates with specific driver events, as tumour subtypes are strongly associated with distinct initiating mutations [36]. *EZH2* and *EMX2* levels significantly anti-correlated in classical, mesenchymal and neural GBMs, suggesting that the multiple genetic drivers may lead to EZH2-mediated *EMX2* repression (Fig EV4B). In addition, both

low and high-grade glioma showed inverse correlation between *EZH2* and *EMX2* mRNA levels (Fig EV4C).

Expression data from laser microdissected regions corroborated the anti-correlation observed across patients, showing opposing *EMX2* and *EZH2* expression patterns between tumour regions and normal adjacent tissue within individual samples (Fig 4D). Supporting an *EMX2-HOX* switch, tumour regions also showed high levels of multiple *HOX* genes from all clusters (Fig EV3C). To further characterise the relationship between *EZH2*, *EMX2* and *HOX* genes in single cells, we performed dual-colour RNA FISH. As expected, *EZH2* mRNA was detected in 4/4 patients, whilst *EMX2* showed low or undetectable expression (Figs 4E–G and EV5A). In the few cells where *EMX2* mRNA was detected, EZH2 levels were low, confirming the inverse relationship between the two genes at the single cell level (Fig 4G). *HOXB9* mRNA was also readily detected in GBM samples, albeit exhibiting some degree of inter- and intra-tumour heterogeneity (Fig EV5B–D). Importantly, repression of *EMX2* and upregulation of many *HOX* genes significantly correlated with tumour grade, indicating the clinical relevance of the aberrant transcriptional patterns observed in patients (Figs 4H, EV3D and EV4D). Altogether, these results strongly support an EZH2-mediated *EMX2-HOX* switch in glioma.

**EZH2-mediated repression of *EMX2* is required for maintenance of tumorigenic potential by glioblastoma cells**

*HOX* genes are established oncogenes that promote tumour development in many tissues when aberrantly expressed [60]. De-repression of *HOX* genes, major EZH2 targets in normal brain cells, is thus a likely mechanism through which the physiological function of EZH2—maintenance of cell identity—is compromised in cancer. However, aberrant expression of *HOX* genes, which are no longer regulated by EZH2 upon transformation, cannot explain the

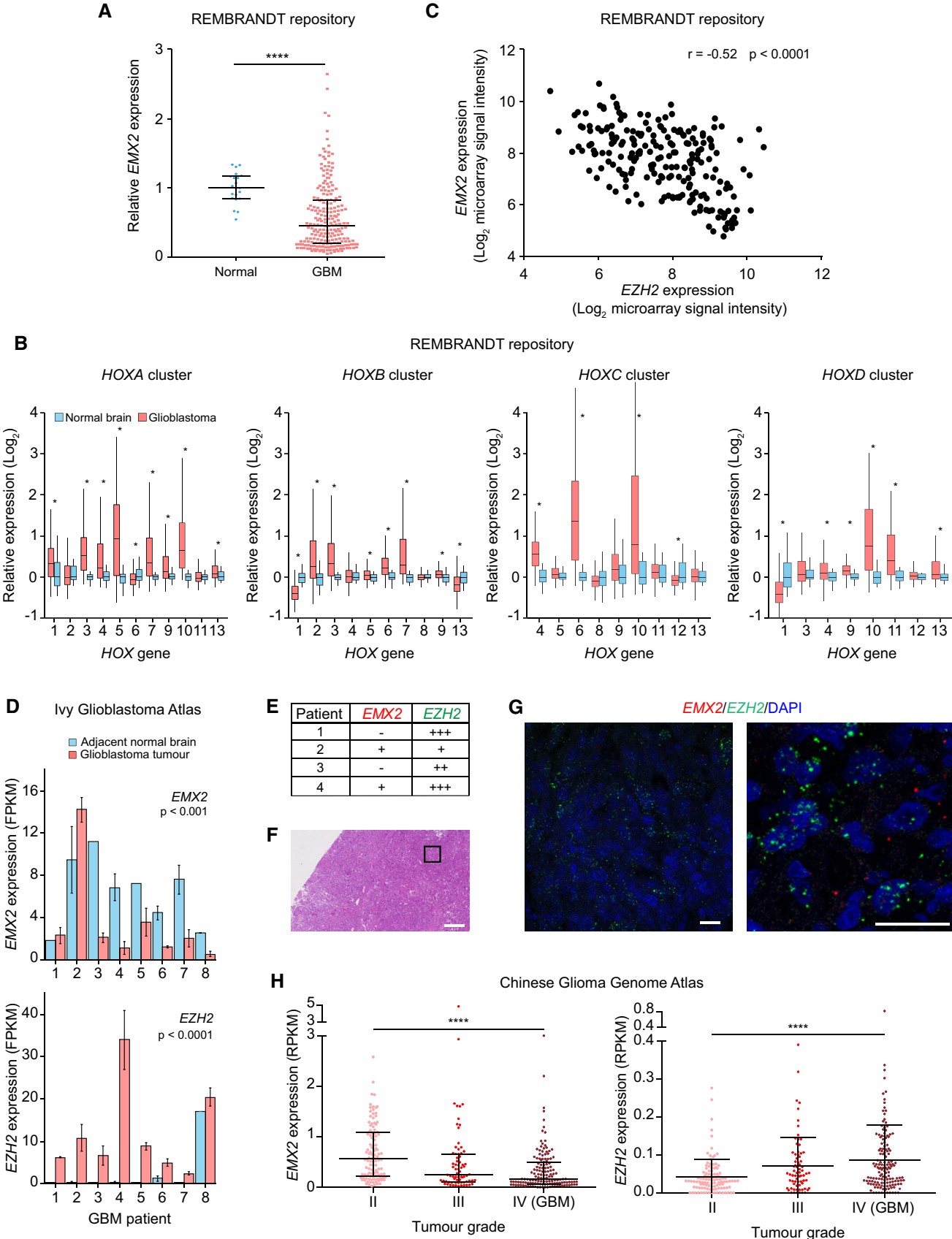

**Figure 4.**

**Figure 4. Aberrant silencing of *EMX2* and de-repression of *HOX* genes in glioma patients.**

A The expression of *EMX2* in normal brain and GBM patient samples. Data sourced from the Repository of Molecular Brain Neoplasia Data (REMBRANDT). Four asterisks indicate *P*-value < 0.0001 (two-tailed Mann–Whitney *U*-test). Bars represent median ± interquartile range. Values are expressed relative to the mean of normal brain samples. *N*: 21 for normal brain, 214 for GBM. Note that the extent of *EMX2* repression in GBM patients is likely underestimated due to the possible presence of normal adjacent tissue in the analysed samples, and due to the intrinsic background noise of microarrays, which limits detection of truly silenced genes.

B Relative expression of *HOX* genes in normal brain and GBM patient samples as detected by microarray analysis. Data sourced from REMBRANDT. Values are expressed relative to the median of normal brain samples. In the boxplots, the top, middle and bottom box delimiters represent the 75th, 50th and 25th percentiles of the data, respectively. Top and bottom whiskers show the 75th percentile + 1.5*interquartile range and 25th percentile − 1.5*interquartile range, respectively. One asterisk indicates *P*-value < 0.05 (two-tailed Mann–Whitney *U*-test corrected for multiple comparison using Holm's method). *N*: 21 for normal brain, 214 for GBM. Data were not available for *HOXD8*.

C Covariance between *EMX2* and *EZH2* expression levels in GBM patient samples, as detected by microarray analysis. Data sourced from REMBRANDT. *P*-value and correlation coefficient (*r*) of the covariance are shown (Spearman rank correlation). Every dot is a patient. *N*: 214.

D The expression levels of *EMX2* (top) and *EZH2* (bottom) in tumour or adjacent normal regions laser microdissected from human GBM tumours as detected by RNA-seq. Data sourced from the Ivy Glioblastoma Atlas. FPKM, fragments per kilobase of transcript per million. The significance of the differential expression in normal and tumour regions across patients is indicated (two-way ANOVA). Bars represent mean ± SEM. *N*: 3 regions sampled for each GBM tumour, 2, 3, 2, 3, 2, 3, 3 and 1 regions for normal tissue of patients 1–8, respectively.

E Scoring of *EMX2* and *EZH2* mRNA staining intensity in human GBM tumour samples. Scoring system: −0 to 0.5 foci/nuclei, +0.51–0.75 foci/nuclei, ++0.76–1 foci/nuclei and +++> 1 foci/nuclei (see Materials and Methods).

F Haematoxylin and eosin (H&E) staining of a human GBM tumour. The black square represents the approximate location of the field shown in (G) (left image). Serial sections were used for H&E and RNA FISH. Scale bar: 300 μm.

G Visualisation of *EZH2* (green) and *EMX2* mRNA (red) in a human GBM tumour by RNA FISH. Nuclei were counterstained with DAPI. Scale bar: 20 μm.

H The expression of *EMX2* (left) and *EZH2* (right) in glioma patient samples, grouped according to tumour grade, as detected by RNA-seq. Data sourced from the Chinese Glioma Genome Atlas. Four asterisks indicate *P*-value < 0.0001 (Kruskal–Wallis test). Bars represent median ± interquartile range. *N*: 109 for grade II, 72 for grade III and 144 for grade IV. RPKM, Reads per kilobase of transcript per million.

pathological function gained by EZH2 in glioma. We therefore focused on *EMX2*, which becomes an EZH2 target specifically in cancer cells. During neurogenesis, EZH2 and EMX2 are co-expressed in NSCs, where they act, respectively, to sustain a proliferative state and restrict cell division [15,55]. We therefore hypothesised that whilst complementary functions of EZH2 and EMX2 ensure controlled self-renewal of normal cells, cancer cells may benefit from *EMX2* silencing to unlock unrestrained proliferation. To test this possibility, we re-expressed *EMX2* at physiological levels (Appendix Fig S1B) in the GBM cell lines U-87 MG and DBTRG-05MG and examined the effect of EMX2 on cell proliferation and tumorigenic potential. Since *EMX2* silencing in these cells depends on EZH2 activity (Fig 3G), this approach assesses the importance of PRC2-mediated *EMX2* repression. The expression of *EMX2* significantly inhibited the proliferation of both GBM cells lines, halving the number of cells in the population after 8 days, whilst RFP used as a control had no effect (Fig 5A). Of note, the slow onset of the effect suggests inhibition of long-term proliferative potential rather than immediate cell cycle arrest, in line with the role of EMX2 in regulating the balance between symmetric and asymmetric divisions [55]. More importantly, re-expression of *EMX2* completely prevented the growth of GBM xenografts in immunocompromised mice, indicating a potent tumour-suppressive function for EMX2 (Fig 5B–D). We conclude that silencing of *EMX2* by EZH2 is required for maintenance of tumorigenic potential by GBM cells and is a major mechanism underpinning the pathological role of EZH2 in glioma.

## Discussion

Increasing evidence suggests that many epigenetic regulators are co-opted by cancer cells to sustain malignant phenotypes such as aberrant proliferation, altered differentiation potential, enhanced resistance to stress and ability to evade immunosurveillance [61–64]. Notably, the pathological function gained by the hijacked proteins in cancer (i.e. *sustaining* aberrant cell behaviour) is antithetic to the role they exert in physiological conditions, where they instead *prevent* abnormal cell behaviour by ensuring maintenance of proper cell identity. Furthermore, epigenetic regulators often acquire a tumour-promoting role in the absence of genetic alterations that affect their molecular properties, indicating that identical proteins exert opposite functions in normal and transformed cells [5,61,62]. Focusing on the Polycomb component EZH2 and its role in the CNS, we show here that underpinning the switch from physiological to pathological function is a genome-wide redistribution of EZH2 induced by oncogenic signalling, and consequent misregulation of key homeotic genes (Fig 6).

Based on the observed overexpression of EZH2 in many cancer types and the correlation with poor patient outcome, the pathological function of EZH2 has generally been attributed to hyperactivation of the PRC2 methyltransferase activity and strengthened repression at existing target genes (e.g. *CDKN2A*/p16) [65]. However, EZH2 overexpression is rarely accompanied by a matched increase in global H3K27me3 levels, and in fact, evidence from genetic studies in both mouse and human systems suggests that EZH2 upregulation in cancer may be a response to cell proliferation and the need to compensate for cell division-induced dilution of H3K27me3 [66–69]. In line with this notion, we show that despite an increase in EZH2 levels upon *de novo* transformation, H3K27me3 levels remain constant across all cellular states. In contrast, EZH2 undergoes extensive redistribution across the genome and generates distinct domains of repressive chromatin in normal and malignant cells. Our results suggest a parallelism between the effect of developmental signalling regulating lineage commitment and oncogenic signalling triggered by driver mutations with respect to PRC2 distribution on chromatin. During differentiation, external cues drive a genome-wide redistribution of PRC2, promoting its dissociation

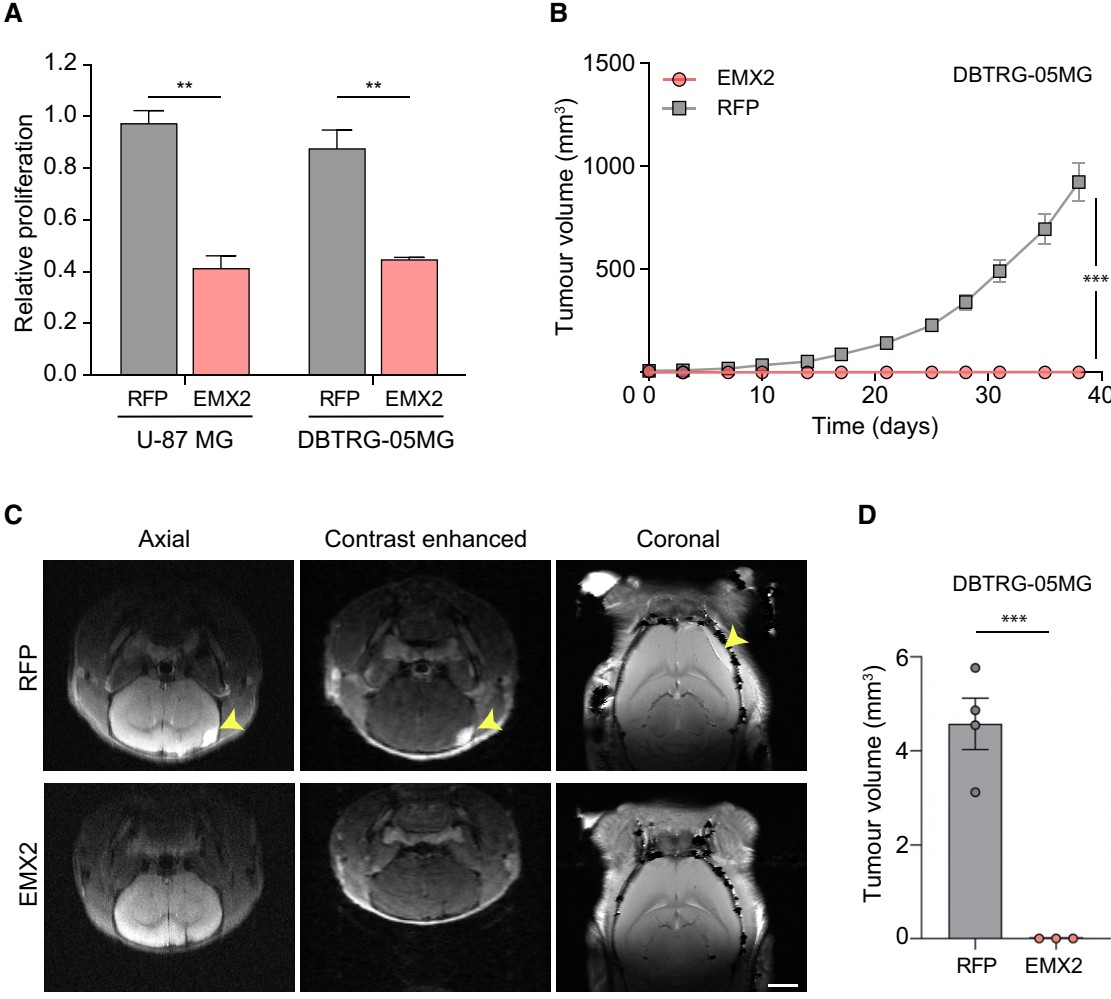

**Figure 5. Re-expression of EMX2 inhibits the tumorigenicity of glioblastoma cells.**

A Proliferation assay examining the effect of *EMX2* ectopic expression in U-87 MG and DBTRG-05MG GBM cells. RFP is used as a control. Values represent mean ± SEM from three biological replicates. The fold change in cell number after 8 days of proliferation whilst expressing the exogenous protein is shown relative to an uninduced control. Two asterisks indicate *P*-value < 0.01 (one-tailed unpaired Student's *t*-test). Similar results were reported in [104] using other GBM cell lines.

B Transplantation assay comparing the growth kinetics of subcutaneous DBTRG-05MG-induced tumours expressing *EMX2* at levels comparable to those expressed in normal cells, or expressing RFP as a control. Values represent mean ± SEM from six tumours. Three asterisks indicate *P*-value < 0.001 (two-tailed single sample Student's *t*-test). Similar results were obtained in an independent experiment using a distinct batch of transduced cells.

C MRI scans showing representative brain tumours 6 weeks after intracranial injection of DBTRG-05MG cells expressing *EMX2* or RFP. Axial anatomical scan and coronal signal intensity map indicate location of tumour in RFP control (yellow arrow). Contrast enhancement in post-contrast images indicates tumour blood–brain barrier breakdown. Signal was not present in pre-contrast images. Images show the same MRI slice position between mice. Scale bar: 2 mm.

D Quantification of brain tumour size in mice injected with *EMX2*- or RFP-expressing DBTRG-05MG cells. Values represent mean ± SEM from four control and three *EMX2* tumours. Five mice per conditions were injected but three (one for RFP and two for *EMX2*) had to be excluded from the study due to complications from the procedure. Three asterisks indicate *P*-value < 0.001 (two-tailed unpaired Student's *t*-test).

from lineage-specifying genes and binding to a distinct set of genes, including those supporting self-renewal and pluripotency/multipotency in stem cells [70–72]. Similarly, we find that oncogenic insults change the set of genes repressed by EZH2, with a critical switch induced by activation of oncogenic RAS signalling that leads to de-repression of tumour-promoting transcription factors and *de novo* silencing of tumour-suppressive ones. Thus, whilst extracellular signalling instructs EZH2 binding in physiological conditions to ensure timely and spatially correct activation of gene expression programmes, oncogenic cell-autonomous mechanisms lead to

aberrant redistribution of PRC2 on chromatin, compromising cell function. It is conceivable that other chromatin regulators involved in cell fate determination may undergo a similar redistribution upon transformation and thereby contribute to the maintenance or enhancement of malignant phenotypes.

Despite extensive EZH2 redistribution across the genome, only 14 direct targets undergo de-repression upon transformation, and at any cellular state, more than 80% of EZH2-bound genes do not respond to EZH2i. This observation is in line with previous reports showing that inhibition of PRC2 function has moderate effects on

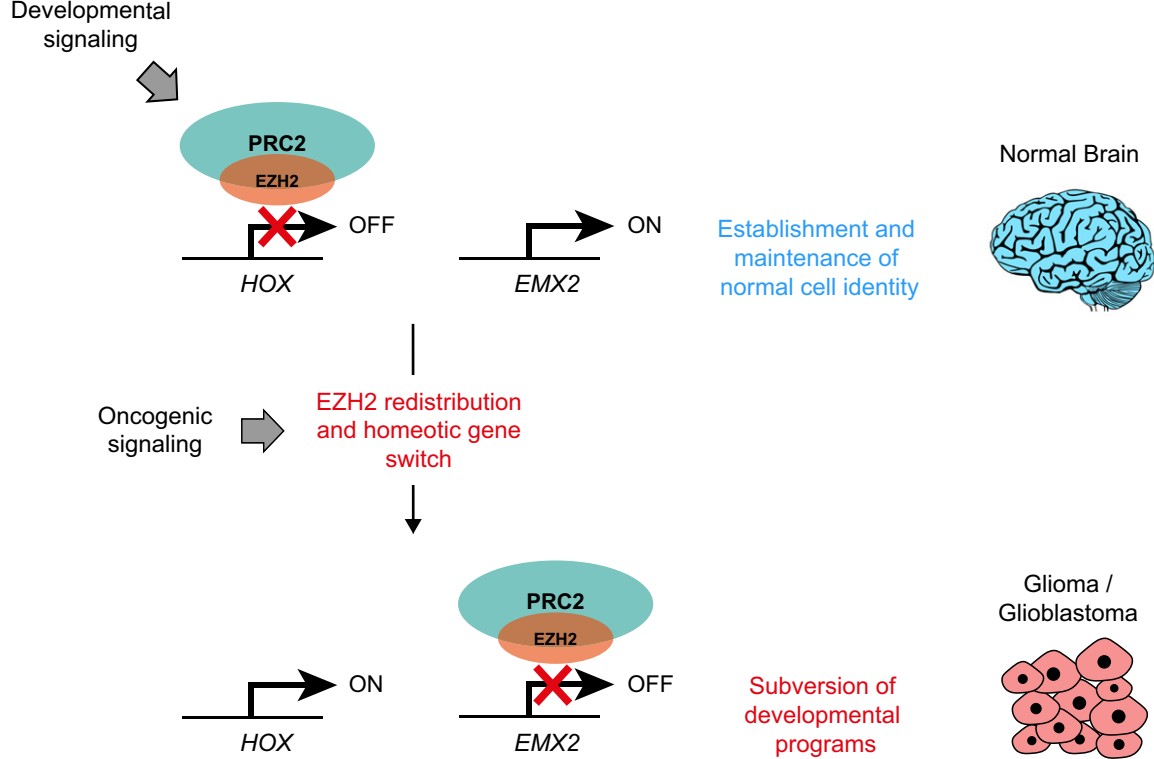

**Figure 6. Model of how neoplastic transformation corrupts the physiological function of EZH2 in glioma.**

During embryonic development, cell-extrinsic cues establish lineage-specific chromatin landscapes which support normal brain function. PRC2 maintains cell identity in the forebrain by repressing spinal cord-specifying *HOX* genes and allowing expression of *EMX2*, a critical regulator of neurogenesis and inhibitor of neural stem cell proliferation. Upon activation of oncogenic signalling by driver mutations, cell-intrinsic changes lead to PRC2 redistribution on chromatin and, as a consequence, to a switch in the expression of homeotic genes. With an altered identity, cells lose physiological brakes that restrain their proliferation and glioma develops. Since maintenance of the rewired transcriptional programmes is required for sustaining malignant cellular properties, glioma cells become dependent on PRC2 and thus vulnerable to EZH2 inhibition.

gene expression [73]. We find that EZH2i-insensitive genes generally show high levels of DNA methylation, suggesting that redundant repressive mechanisms may be acting at a subset of PRC2-bound genes in cancer cells. Our findings are in agreement with the observation that EZH2 and the DNA methyltransferase DNMT1 interact with each other and co-operate in silencing genes in U2OS sarcoma cells [74]. Interestingly, H3K27me3 and DNA methylation distributions strongly anticorrelate in embryonic stem cells [75,76], suggesting that the functional relationship between PRC2-related chromatin domains and DNA methylation may differ in normal and cancerous cells. An additional explanation for the low fraction of EZH2 target genes responding to EZH2i is that loss of repressive chromatin may not be sufficient to transcribe a gene if relevant transcription factors needed to activate transcription are not present.

Amongst the few genes that respond to EZH2 redistribution and show aberrant expression in glioma, we find key homeotic genes that specify cell fate in distinct regions of the CNS. De-repression of *HOX* genes, classical PRC2 targets, has been observed in various cancers, and compelling evidence indicates a tumour-promoting role for these proteins when aberrantly expressed in adult tissues [77–79]. Although our characterisation of *de novo* transformed cells identified *HOXB9* as primarily affected by EZH2 redistribution, we observed loss of H3K27me3 domains from multiple *HOX* clusters,

especially at more posterior genes (*HOX6-HOX13*). Furthermore, aberrant expression of numerous *HOX* genes was detected in GBM cell lines and patients, indicating that repressive chromatin at *HOX* clusters is generally destabilised in cancer cells. A more surprising and intriguing finding is the *de novo* repression of *EMX2* by EZH2 upon transformation. EMX2 is mostly known for its role in the developing forebrain, where it is required for timely formation of the dentate gyrus, the medial limbic cortex and the olfactory bulbs [80]. EMX2 continues to be expressed in the periventricular region of the adult brain, where it acts as a negative regulator of NSC proliferation by favouring asymmetric cell division [55]. Interestingly, EZH2, which is also expressed in NSCs, regulates stem cell self-renewal in the opposite way, by inhibiting differentiation and maintaining a proliferative state [15]. Thus, the concerted action of EZH2 and EMX2 may control the balance between self-renewal and differentiation in the neurogenic area of the adult brain and ensures proper tissue maintenance. In contrast, we show that *EMX2* becomes a direct target of EZH2 upon transformation and is broadly silenced in GBM patients. Notably, recent work shows that GBM likely originates in NSCs of the periventricular region, supporting the relevance of such a mechanism in human GBM [81]. We speculate that transformed cells, either mutated NSCs or committed cells which have been reprogrammed to a more undifferentiated state by oncogenic insults, may hijack EZH2 to silence its antagonist and

thereby unleash uncontrolled self-renewal. In agreement with this model, forced expression of EMX2 in GBM cells completely prevents tumour formation, indicating the necessity for GBM cells to stably silence *EMX2* to maintain tumourigenic potential.

Our results provide an explanation for the paradoxical dual role of EZH2 and PRC2 in cancer. In addition to being critical for the maintenance of various cancers, EZH2 has also been shown to exert a tumour-suppressive function: loss-of-function mutations in many PRC2 members are prevalent in various cancers, and mouse models deficient for EZH2 or other PRC2 components show cancer predisposition [18,82]. This dual role has been attributed to the fact that PRC2 may have tissue-specific functions and/or respond differently depending on the genetic drivers that initiate the disease [65]. We propose that EZH2 acts as a tumour suppressor in all normal or premalignant cells, where it exerts its physiological function and cooperates with other epigenetic regulators to maintain proper cell identity in the face of cell-extrinsic or intrinsic perturbations. If EZH2 or other PRC2 components are lost in these cells due to mutations, oncogenic insults can induce aberrant cell behaviour more easily and cancer development is favoured. In this scenario, EZH2 therefore acts as a tumour suppressor in the earliest stages of tumorigenesis. However, after cells transform, as a consequence of the cellular changes induced by oncogenic signalling, EZH2 undergoes a redistribution on chromatin and by repressing genes that inhibit malignant phenotypes, such as *EMX2*, it acquires a pathological function. The apparently conflicting functions of EZH2 in cancer may therefore simply reflect the antithetic roles that the protein plays at distinct stages of the disease: tumour suppressor during tumour-initiation, and tumour-promoter after cells transform and reprogramme their epigenome. Of note, recent findings support this model in acute myeloid leukaemia (AML) as well [83]. Thus, the stage-specific role of EZH2 in cancer may be a widespread mechanism, which influences the evolution of both solid and haematological cancers.

# Materials and Methods

### Cell lines and constructs

All cell lines used in this study were cultured at 37°C in 5% $CO_2$ using the media conditions stated in Appendix Table S1. For cell lines transduced with pTRIPZ doxycycline-inducible constructs, tetracycline-free foetal bovine serum was used to supplement the media to prevent undesired expression from the pTRIPZ construct. All cell lines were sourced as stated in Appendix Table S1 and subsequently tested by STR profiling and validated as mycoplasma free.

For inducible cDNA expression, pTRIPZ (Dharmacon) was modified to introduce an SV40-poly A signal and blasticidin resistance as previously described [33]. EMX2 cDNA (coding sequence of NM_004098.3) was amplified by PCR from a Precision LentiORF EMX2 plasmid (Dharmacon) and subcloned into modified pTRIPZ using AgeI-BstBI sites. For all cDNA overexpression experiments, empty pTRIPZ, expressing puromycin resistance, mir30 cassette, rtTA3 and TurboRFP, was used as negative control.

For CRISPR-Cas9-mediated knock-out of *EZH2*, an *EZH2*-targeting sgRNA (5′-ACACGCTTCCGCCAACAAAC-3′) was cloned into pLENTI_GFP_sgRNA as previously described [84]. The sgRNA sequence was selected using the MIT sgRNA design tool (crispr.mit.edu) as the top hit against EZH2's first exon.

To generate lentivirus, HEK293T cells were transfected with pMD2G, psPAX2 and the construct of interest using FugeneHD (Promega). After 24 h, virus was harvested and diluted 1:1 in the appropriate media plus 8 μg/ml polybrene (Merck Millipore) then applied to the cell line. After a further 24 h, the media was changed and selection was initiated. For pTRIPZ constructs, 1 μg/ml puromycin or 5 μg/ml blasticidin was used to select both DBTRG-05MG and U-87 MG lines. pLENTI_GFP_sgRNA constructs were selected by fluorescence-activated cell sorting (FACS) of GFP-positive cells.

For *EZH2* knock-out experiments, a transformed fibroblast cell line expressing inducible humanised-Cas9 [84] was transduced with either an EZH2 or an eGFP targeting sgRNA. After FACS of GFP-positive cells, the polyclonal transduced population was induced with 1 μg/ml doxycycline for 21 days to induce *EZH2* KO and consequent loss of H3K27me3. CRISPR-Cas9 editing at the EZH2 locus was assessed by TIDE analysis [85] and Western blot.

### Proliferation assays

U-87 MG and DBTRG-05MG cell lines expressing inducible RFP or EMX2 were pre-induced with 1 μg/ml doxycycline for 7 days, subsequently plated at a density of 10,000 cells/well in triplicate on a 6-well plate and compared with corresponding uninduced cells. After 16 h, the plates were phase imaged using an IncuCyte S3 (Essen bio) to allow time zero normalisation of cell plating. Cells were then grown ±1 μg/ml doxycycline for 8 days. To quantify the endpoint cell number, plates were stained with SYTOX (ThermoFisher) and nuclei were counted using IncuCyte image analysis software. The endpoint values were then normalised relative to time zero based on the object count calculated from the initial phase images.

### Protein immunodetection

Protein was extracted from cultured cells by resuspension in high salt buffer (50 mM Tris pH 7.5, 300 mM NaCl, 0.5% IgePal and 1 mM EDTA) followed by three cycles of sonication (30 s on/off) using a chilled Bioruptor Pico sonicator (diagenode). Protein levels were quantified by Bradford assay, and the samples were boiled for 7 min in LDS sample buffer (NuPAGE) and reducing agent (NuPAGE). Samples were run on a 4–12% bis-tris gel (Life Technologies) and transferred using a Life technologies iBlot2 system. Membranes were blocked with 0.1% Tween 20 in PBS (PBT) + 5% milk and then blotted with the appropriate primary antibody at the relevant concentration for 1 h at room temperature (RT): EZH2 (Cell Signalling - 5246S, 1:1,000), H3K27me3 (Upstate - 07-449, 1:5,000), Histone H3 (Abcam - ab1791, 1:40,000). The membrane was then washed three times for 10 min at RT in PBT, blotted with an anti-rabbit horseradish peroxidase secondary antibody (Vector - PI-1000, 1:5,000), washed again and then developed using ECL Western blotting substrate (Pierce).

### DNA methylation analysis

Genomic DNA was extracted from transformed fibroblasts using a DNA blood and tissue kit (Qiagen) and bisulphite converted using

the EZ DNA methylation-Direct Kit (Zymo research) in accordance with the manufacturer's protocol. Primers specific to bisulphite-treated DNA (Appendix Table S2) were designed using MethPrimer [86]. Regions were then amplified by PCR using the bisulphite-treated genomic DNA as a template and cloned into pCR 2.1 Topo vector using TOPO TA Cloning (Invitrogen). For each region, individual colonies were sequenced and the resulting data were analysed using QUMA [87].

### In vivo tumorigenicity assays

Tumour studies were performed using 5- to 6-week-old male NOD.Cg-$Prkdc^{scid}$ $Il2rg^{tm1Wjl}$/SzJ (NSG) mice, sourced from the Francis Crick Institute common colony and housed in individually ventilated cages. Intradermal injection was performed following procedures described previously [38]. Briefly, 350,000 DBTRG-05MG cells, expressing either *EMX2* cDNA or an RFP expressing control, were injected intradermally into both flanks of three 5-week male NSG mice for each condition. After appearance, tumour size was measured weekly using digital callipers and volume was calculated as $L*W^2/2$, where $L$ = longest edge of tumour and $W$ = shortest edge of tumour. For orthotropic brain tumours, NSG mice were anesthetised, and upon surgical exposure, a syringe needle was inserted into the striatum (1.5 mm lateral of the bregma, 2.5 mm deep) [88]. 300,000 DBTRG-05MG cells expressing either *EMX2* cDNA or an RFP expressing control were delivered to five mice for each condition. Three mice (one RFP control and two EMX2) were removed from the study early from health complications, with no tumours evident at endpoint. Tumour appearance was monitored by magnetic resonance imaging (MRI) from 3 weeks. Tumour volume was measured from MRI anatomical axial scans using ImageJ software and calculated as the average area of tumours in $mm^2$ from consecutive sections multiplied by the depth (0.32-mm section interval multiplied by number of sections). Mice were randomly allocated for injection, ensuring that animals of similar age were present in both conditions. Assessment of results was not performed blinding the investigator. Animal studies were conducted in accordance with the Francis Crick Institute project licence PPL 70/8167 approved by the Home Office.

### Magnetic resonance imaging (MRI)

MRI scans were performed on a 9.4T 20 cm bore MRI scanner (Bruker Biospec; Ettlingen Germany) equipped with a 4-channel mouse head array r.f. receive coil and a 86 mm volume r.f. transmit coil. Paravision 6.0.1 software (Bruker; Ettlingen Germany) was used for acquisition and T1 fitting. Localiser scans were used for consistent placement of slices. T2-weighted (T2W) anatomical scans using a rapid acquisition with refocussed echoes (RARE) sequence were performed with the following scan parameters: RARE factor 8, effective echo time (TE) = 12 ms and echo spacing = 12 ms, repetition time (TR) = 2,562 ms, 24 × 0.32 mm axial slices, FoV 20 × 20 mm, 256 × 192 acquisition matrix zero-filled and reconstructed with a 256 × 256 matrix, 4 averages. T1 maps were acquired using a variable repetition time RARE protocol with the following parameters: RARE factor 2, effective TE = 8 ms and echo spacing = 8 ms, TR = 330, 662, 1,088, 1,686, 2,795, 7,500 ms, 14 × 0.5 mm coronal slices, FoV 20 × 20 mm, 256 × 192 matrix.

Contrast enhanced scans were performed using a fast low angle shot (FLASH) protocols with the following parameters: TE = 2.1 ms, TR = 140 ms, 50° flip angle, 66 × 128 acquisition matrix zero-filled and reconstructed with a 128 × 128 matrix, identical slice position and orientation to the anatomical scans, and 120 repetitions with a time resolution of approx. 18 s. 30 μl of 0.5 molar dimeglumine gadopentetate (Magnevist, Schering, Berlin, Germany) was injected intravenously via a tail-vein cannula approximately 5 min after the start of the contrast enhanced scans [89]. Isoflurane anaesthetic (1–2%) in oxygen [enriched air] and a heated pad was used to maintain core temperature and respiration rate of the mice throughout all scans.

### Chromatin immunoprecipitation (ChIP)

ChIP was performed for EZH2 and H3K27me3 in an identical manner for all cell lines profiled in this study. For each line, 20 million cells were fixed with 30 ml 1% formaldehyde in cell culture media for 10 min at RT. The fixation was quenched by addition of 125 mM glycine for a further 5 min. The fixed cells were washed twice in ice-cold PBS and resuspended in 1.8 ml 1:1 SDS-containing buffer (100 mM NaCl, 50 mM Tris–HCl pH 8.0, 5 mM EDTA pH 8.0, 0.2% NaN₃, 0.5% SDS): triton-containing dilution buffer (100 mM NaCl, 500 mM Tris–HCl pH 8.6, 5 mM EDTA pH 8.0, 0.2% NaN₃, 5% Triton X-100) plus protease inhibitors (Cell Signalling). The suspension was incubated on ice for 20 min. Chromatin was sheared to 200–400 bp using a Bioruptor Pico Sonicator (Diagenode), with 15 cycles of 30 s on/off. The resulting lysate was clarified by 30 min of centrifugation at 10,000 × $g$ and quantified by Bradford assay. EZH2 (Cell Signalling - 5246S, 1:40), H3K27me3 (Upstate - 07-449, 1:100) or Rabbit IgG control antibodies (Abcam - ab46540, 1:100) were combined with 1 mg of chromatin lysate, made up to 1 ml with IP buffer (1:1, SDS:Triton buffers) and rotated at 4°C overnight. All immunoprecipitations were performed in duplicate using the same chromatin lysate. The following day, 30 μl of protein-G magnetic beads (ThermoFisher) were added to the immunoprecipitations and rotated at 4°C for 4 h. The bead-antibody complexes were washed three times with ice-cold wash buffer 1 (1% Triton X-100, 0.1% SDS, 150 mM NaCl, 2 mM EDTA pH 8.0, 20 mM Tris–HCl) and once with ice-cold wash buffer 2 (wash buffer 1 with 500 mM NaCl). Chromatin was eluted from the beads by shaking at 65°C overnight with 110 μl of 0.1 M NaHCO₃ 1% SDS solution. DNA was isolated from the eluate by PCR purification (Qiagen). For ChIP-seq samples, a quality control ChIP-qPCR was performed using primers (Appendix Table S2) against two positive control regions within the *WT1* gene and a negative control region in *GAPDH*. Positive and negative controls were selected based on publically available ChIP-seq datasets. The quantity and integrity of the chromatin were assessed using a BioAnalyser 2100, and library preparation was undertaken with 5–10 ng of DNA using the Illumina TruSeq ChIP protocol.

### ChIP-seq analysis

ChIP-seq samples were multiplexed five or six per lane and sequenced using an Illumina HiSeq 2500 producing 101 bp paired-end reads. For the Transformed_H3K27me3_rep1 sample, two batches of immuno-precipitated chromatin were sequenced, and the resulting FASTQC

files were merged to achieve sufficient coverage. Sequencing run quality was assessed using FastQC (Andrews S, 2010). Adapter trimming was performed with cutadapt (version 1.9.1) [90] with parameters "–minimum-length = 25 –qualitycutoff = 20 -a AGATCGGAAG AGC -A AGATCGGAAGAGC". BWA (version 0.6.2) [91] with default parameters was used to perform genome-wide mapping of the adapter-trimmed reads to the human hg19 genome. Duplicate marking was performed using the picard tool MarkDuplicates (version 2.1.1; Broad Institute), and duplicate reads were subsequently removed. Alignments were then filtered to remove reads that mapped to DAC Blacklisted Regions from the ENCODE/DAC [92] downloaded from the UCSC. Further filtering was performed to exclude read pairs that were discordant, mapped to different chromosomes, ambiguously mapped, and had a mismatch > 4 in any read. Tiled data format (tdf) files for ChIP-seq visualisation were produced using the "count" function in IGVTools (version 2.3.75) with default parameters. For compatibility with many downstream applications, it was necessary to obtain fragment-level intervals for each paired-end read. To achieve this, bam files were first converted to bed paired-end format using the bamtobed function in BEDTools (version 2.26 in all instances) [93]; the BED files were then generated by keeping the furthest extent of both paired-end reads. Peaks were called on each replicate for all samples using SICER (version 1.1) [94] with the following parameters: "redundancy threshold = 1, window size = 200, fragment size = 110, effective genome fraction = 0.75, gap size = 400 or 600 (EZH2 and H3K27me3, respectively) and FDR < 0.0001". A consensus set of peaks for each cellular state was derived by merging replicate peaks using "mergePeaks" from the homer package (version 4.8.3 in all instances) [95] and extracting the summed overlapping peaks. Resulting peaks separated by < 250 bp were merged into a single peak using "mergeBed" from BEDTools. EZH2 and H3K27me3 peak sets for each cellular state were intersected, and only EZH2 peaks overlapping a H3K27me3 peak for > 25% of their width were retained. Consensus EZH2 peak sets from each cellular state were intersected with "mergePeaks" (HOMER), giving cellular state unique and common peaks, from which Venn diagrams of EZH2 binding distribution were generated. To calculate tag enrichment at these peak sets, tag directories were first compiled for the combined reads of both EZH2 ChIP-seq replicates from each cellular state using "makeTagDirectory" (HOMER) with default parameters. Relative enrichment was then calculated at all common and unique EZH2 peaks using "getDifferentialPeaks" (HOMER) with the following parameters set to non-default values: "-F = 0, -P = 1, -tagAdjust = 0, -tagAdjustBg = 0". EZH2 peaks with a fold change ≥ 1.5 and a $P$-value ≤ 1e-20 ("large-magnitude" differential peaks) relative to the compared cellular state were then considered for further analysis. A third replicate of EZH2 ChIP-seq generated and processed as indicated above was used to validate differential regions identified using the first two replicates. The selected peaks were annotated to the nearest transcription start sites (TSS) of a protein coding, antisense or lincRNA gene using the Refseq hg19 TSS annotation. If a peak overlapped with, or was equidistant from, multiple TSSs, then all TSSs were recorded. Only those peaks found within 5 kb of a TSS were considered for further analysis. Overlaps between the gene set associated (±5 kb TSS) with large-magnitude differential peaks and existing gene signatures were calculated using "compute overlaps" from the Broad Institute (software.broadinstitute.org/gsea/msigdb/annotate.jsp).

ChIP-seq metaprofiles were plotted with ngsplot (version 2.63) [96] using the following parameters: "normalisation = bin, colour scaling = global and a fragment length = 300".

CpG island annotations were sourced from the UCSC and are based on the epigenomic predictions of Bock et al [97] (http://hgdownload.soe.ucsc.edu/goldenPath/hg19/database/cpgIslandExt.txt.gz).

Lamin-associated domain boundaries in human fibroblasts were sourced from Guelen et al [45] (https://www.ncbi.nlm.nih.gov/geo/query/acc.cgi?acc = GSE8854).

## RNA-seq and reverse transcription quantitative PCR (RT–qPCR)

RNA was extracted from all cell lines analysed using an RNeasy Plus Mini Kit (Qiagen) in accordance with the manufacturer's instructions. For RNA-seq, RNA integrity was assessed using a TapeStation 4200 (Agilent) and all samples were found to have an RNA integrity number ≥ 8. Libraries for sequencing were prepared using the KAPA Stranded RNA-Seq Kit with RiboErase (Roche) according to the manufacturer's instructions. Library quality was confirmed using a BioAnalyser 2100. For RT–qPCR, 1 μg of RNA was reverse transcribed using a High Capacity cDNA Reverse Transcription Kit (ThermoFisher) as per the manufacturer's instructions. RT–qPCR was then performed using SsoAdvanced™ Universal SYBR® Green Supermix (Bio-Rad) on a CFX96 real-time PCR detection system (Bio-Rad). For all experiments, the housekeeping gene cyclophilin A (*PPIA*) was used as reference. Primers used for RT–qPCR in this study can be found in Appendix Table S2.

For experiments involving EZH2 pharmacological inhibition, cell lines were treated with either EPZ-6438 (Selleckchem), dissolved in DMSO, or DMSO alone for 12 days. To maintain repression throughout the treatment period, drug and media were replaced every 3 days.

## RNA-seq analysis

RNA sequencing was carried out on the Illumina HiSeq 4000 platform and typically generated ~20 million 75 bp single-end reads per sample. To reduce sequencing lane biases, the library of each sample was split across two lanes, generating two fastq files for each sample. These were subsequently merged before the downstream analysis. The resulting reads were adapter-trimmed using cutadapt as specified previously. The RSEM package (version 1.2.29) [98] in conjunction with the STAR alignment algorithm (version 2.5.1b) [99] was used for the mapping and subsequent gene-level counting of the sequenced reads with respect to hg19 RefSeq genes downloaded from the UCSC Table Browser [100] on 14th April 2016. The parameters used were "–star-output-genome-bam –forward-prob 0". Differential expression analysis was performed for DMSO vs. EPZ-6438 (EZH2 pharmacological inhibitor)-treated cells with the DESeq2 package (version 1.10.1) [101] within the R programming environment (version 3.2.3) [102]. Differentially expressed genes (DEGs) were then defined as those with an FDR ≤ 0.01, $Log_2FC$ ≥ 1 and maximal TPM across conditions greater than 1. The DEGs from each condition were intersected and separated into those regulated by EZH2 in a specific condition or across multiple conditions. DEGs, upregulated by inhibition in a condition-specific manner, were intersected with genes associated

with a differential EZH2 peak in the same condition; the overlapping genes were then used for further analysis. Genes differentially expressed between conditions were identified by comparing expression between DMSO-treated conditions using DESeq2.

## Analysis of publically available datasets

### Neural cells and glioma cell lines

RNA-seq mRNA expression values for human primary neural cells were downloaded from the Brainseq2 data portal (http://www.brainrnaseq.org/) [103]. RNA-seq mRNA expression and reduced representation bisulphite sequencing DNA methylation values from glioma cell lines were downloaded from the Cancer Cell Line Encyclopaedia data portal (https://portals.broadinstitute.org/ccle/data). Analysis of DNA methylation at EZH2i sensitive and insensitive genes was performed as follows: processed gene-level methylation data for all NCI-60 cell lines were downloaded from National Cancer Institute via the cell miner database (discover.nci.nih.gov/cellminer/) and genes lacking methylation data in any cell line were excluded. Data for genes with an EZH2 peak ±5 kb TSS, as identified by ChIP-seq, were selected for the analysis. Genes with an EZH2 peak were further subdivided into genes sensitive/insensitive to EZH2i as identified by RNA-seq analysis. Sensitive genes were defined as those upregulated with a $Log_2FC \geq 1$, $FDR \leq 0.01$ and a maximal TPM in the relevant cellular state greater than 1 upon EZH2i treatment.

### Patient samples

For patient analysis, multiple datasets which include complementary information were used. The Repository of Molecular Brain Neoplasia Data (REMBRANDT) [59], which contains normal brain samples as controls, was used to assess *EZH2/EMX2/HOX* gene misregulation in GBM patients. Normalised microarray mRNA expression values were obtained via the Betastasis cancer browser (http://www.betastasis.com/glioma/rembrandt/). Analysis of molecular subtypes was performed using The Cancer Genome Atlas (TCGA) GBM dataset, which includes patient classification into classical, mesenchymal, neural and proneural GBM. Normalised microarray intensity values were obtained from via the Betastasis cancer browser (http://www.betastasis.com/glioma/tcga_gbm/). When comparing *EMX2* and *EZH2* levels across samples of different grade [low-grade glioma (LGG) vs. GBM] in TCGA, the RNA-seq dataset, obtained via the UCSC Xena data portal (https://xenabrowser.net/datapages/), was used. The Chinese Glioma Genome Atlas RNA-seq dataset, which contains tumour grade for all glioma samples, was used for comprehensive analysis of *EMX2* and *EZH2* levels in grade II, III and IV gliomas. Data were accessed via the data portal GLIOMASdb (http://cgga.org.cn:9091/gliomasdb/download.jsp). RNA-seq data from laser microdissected GBM tumours were obtained from the Ivy Glioblastoma Atlas data portal (http://glioblastoma.alleninstitute.org/rnaseq/search/index.html) and used to compare *EZH2/EMX2/HOX* gene levels in individual patients.

### Human tissue samples

GBM tissue sections from primary resections of adult patient tumours were obtained from Prof. Sebastian Brandner at University College London Hospital with relevant ethical consent provided by

BRAIN UK (Ref: 18/008). Patients had not been treated with chemotherapy prior to surgery.

### RNA *in situ* hybridisation

Formalin-fixed paraffin-embedded GBM tissue sections were stained using an RNAscope® Multiplex Fluorescent Reagent Kit v2 in accordance with the manufacturer's instructions. For staining, the following probes were used: Hs-HOXB9 (473521), Hs-EMX2 (320269-C2) and Hs-EZH2 (405491). As a positive control for EMX2 expression, a cell pellet containing a 1:1 mixture of U-87 MG overexpressing EMX2 and HEC59 endometrial cancer cells endogenously expressing EMX2 was paraffin embedded and sections were stained for EMX2 in parallel to GBM samples. Similarly, as a positive control for HOXB9 expression, a cell pellet containing a 1:3 mixture of transformed fibroblasts overexpressing HOXB9 and PC9 lung adenocarcinoma cells was used. The cell pellet used as a positive control for EMX2 was used as a negative control for HOXB9 staining, whilst the HOXB9-positive control cell pellet was used as a negative control for EMX2 staining. For experiments involving GBM samples, a serial section was stained with haematoxylin and eosin (H&E) to enable identification of the morphological features present. For each sample, 10 images were taken from across the tumour section using a Zeiss 710 confocal microscope. The "Analyse particles" function in ImageJ was used to calculate the number of fluorescent foci present in each image, and the foci count was normalised to the number of nuclei in each image. In Fig 4E, patient samples were scored as −, +, ++ and +++ if they had an average of 0–0.5, 0.51–0.75, 0.76–1 or > 1 fluorescent foci/nuclei across the 10 images, respectively. In Fig EV5D, fields were considered positive if they had > 0.8 foci/nuclei.

### Statistical analysis

Sample size for each experiment was chosen based on estimates of the experimental and biological variability derived from either pilot experiments or similar experiments carried out previously. Unless otherwise stated, all error bars represent ± standard error of the mean for the number of replicates indicated by *N* in the relevant figure legend. All statistical tests used are indicated in the appropriate figure legends. Normality test was automatically performed by the statistical software used to assess significance. If the test failed, an appropriate test was performed. In most cases, the variance was similar between the groups that were being compared. When analysing patient datasets, some of the groups had substantially different *N* (e.g. normal brain and glioma samples), but this was taken into account when assessing the statistical significance of the differences. In boxplots, the top, middle and bottom box delimiters represent the 75th, 50th and 25th percentiles of the data, respectively. Top and bottom whiskers show the 75th percentile + 1.5*interquartile range and 25th percentile – 1.5*interquartile range, respectively.

## Data availability

All ChIP-seq and RNA-seq datasets generated in this study have been deposited in the Gene Expression Omnibus repository with the codes GSE126396 https://www.ncbi.nlm.nih.gov/geo/query/acc.c

gi?acc = GSE126396 and GSE126395 https://www.ncbi.nlm.nih.gov/geo/query/acc.cgi?acc = GSE126395, respectively.

Expanded View for this article is available online.

## Acknowledgements

We thank the Crick Advanced Sequencing for preparing and sequencing NGS libraries, the Biological Research Facility for help with animal work, the Crick Flow Cytometry for help with cell sorting and the Crick Experimental Histopathology for advice regarding RNAscope. Human tissue was obtained from University College London NHS Foundation Trust as part of the UK Brain Archive Information Network (BRAIN UK) which is funded by the Medical Research Council and Brain Tumour Research. BRAIN UK reference number: 18/008—Rewiring of developmental programmes by wild-type EZH2 in cancer cells. This work was supported by The Francis Crick Institute, which receives its core funding from Cancer Research UK (FC001152), the UK Medical Research Council (FC001152) and the Wellcome Trust (FC001152). ZJ and SB are supported by the National Institute for Health Research to UCLH Biomedical Research Centre's funding scheme.

## Author contributions

TM performed most experiments and the downstream computational analysis. ENW performed orthotopic transplantation assays, BMS provided support with MRI, HP performed initial QC and processing of ChIP-seq datasets and the RNA-seq analysis. ZJ and SB provided GBM clinical samples. TM and PS designed experiments, interpreted the results and wrote the manuscript. PS conceived and supervised the study.

## Conflict of interest

The authors declare that they have no conflict of interest.

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
