## [Review Process File · EMBO Reports]

Redistribution of EZH2 promotes malignant phenotypes by rewiring developmental programs

Thomas Mortimer, Elanor N. Wainwright, Harshil Patel, Bernard M. Siow, Zane Jaunmuktane, Sebastian Brandner and Paola Scaffidi

Review timeline:	Submission date:	22 March 2019
	Editorial Decision:	25 March 2019
	Revision received:	15 July 2019
	Editorial Decision:	29 July 2019
	Revision received:	5 August 2019
	Accepted:	8 August 2019

Editor: Achim Breiling

Transaction Report: This manuscript was transferred to *EMBO reports* following peer review at *The EMBO Journal*

1st Editorial Decision

25 March 2019

Thank you for the transfer of your research manuscript to EMBO reports. I have now read your paper and went through the referee reports from The EMBO Journal (which you will find attached at the end of this message).

All referees acknowledge the potential interest of the findings. Nevertheless, all referees have raised a number of concerns and suggestions to improve the manuscript, or to strengthen the data and the conclusions drawn. As the reports are below, I will not detail them here.

Although referee #2 is rather negative, and states that the results are over interpreted and the study lacks proper models and controls, the other two referees are more positive, and indicate that the work does describe an important role of Ezh2 in oncogenic transformation, providing essential mechanistic/molecular insight and proof of concept. Thus, I would be happy to receive a revised version of the manuscript, addressing the concerns of referees #1 and #3, and also toning down the text regarding the model as a true cancer model. In case you also have data to address the points of referee #2, or to strengthen data criticized by this referee, please add these also to the revised manuscript. However, to repeat your experiments in another model system, as indicated by this referee, will not be needed to allow publication of your manuscript in EMBO reports.

Given the constructive referee comments, we would like to invite you to revise your manuscript for EMBO reports with the understanding that the referee concerns must be addressed in the revised manuscript (as indicated above) and in a detailed point-by-point response. Acceptance of your manuscript will depend on a positive outcome of a second round of review. It is our policy to allow a single round of revision only and acceptance or rejection of the manuscript will therefore depend on the completeness of your responses included in the next, final version of the manuscript.

Revised manuscripts should be submitted within three months of a request for revision; they will otherwise be treated as new submissions. Please contact us if a 3-months time frame is not sufficient for the revisions so that we can discuss the revisions further.

Please refer to our guidelines for preparing your revised manuscript and the figure panels:

<http://embor.embopress.org/authorguide#manuscriptpreparation>

http://embopress.org/sites/default/files/EMBOPress_Figure_Guidelines_061115.pdf

Supplementary/additional data: The Expanded View format, which will be displayed in the main HTML of the paper in a collapsible format, has replaced the Supplementary information. You can submit up to 5 images as Expanded View. Please follow the nomenclature Figure EV1, Figure EV2 etc. The figure legend for these should be included in the main manuscript document file in a section called Expanded View Figure Legends after the main Figure Legends section. Additional Supplementary material should be supplied as a single pdf labeled Appendix. The Appendix includes a table of content on the first page, all figures and their legends. Please follow the nomenclature Appendix Figure Sx throughout the text and also label the figures according to this nomenclature. For more details please refer to our guide to authors.

Important: All materials and methods should be included in the main manuscript file.

Regarding data quantification and statistics, can you please specify, where applicable, the number "n" for how many independent experiments (biological replicates) were performed, the bars and error bars (e.g. SEM, SD) and the test used to calculate p-values in the respective figure legends. Please provide statistical testing where applicable. See:
<http://embor.embopress.org/authorguide#statisticalanalysis>

Please also follow our guidelines for the use of living organisms, and the respective reporting guidelines: <http://embor.embopress.org/authorguide#livingorganisms>

We now strongly encourage the publication of original source data with the aim of making primary data more accessible and transparent to the reader. The source data will be published in a separate source data file online along with the accepted manuscript and will be linked to the relevant figure. If you would like to use this opportunity, please submit the source data (for example scans of entire gels or blots, data points of graphs in an excel sheet, additional images, etc.) of your key experiments together with the revised manuscript. Please include size markers for scans of entire gels, label the scans with figure and panel number. Please send one PDF file per figure.

Please add up to 5 key words, and a short running title (up to 40 characters, including spaces) to the title page of the manuscript.

Finally, please format the references according to our journal style. See:
<http://embor.embopress.org/authorguide#referencesformat>

- a complete author checklist, which you can download from our author guidelines (<http://embor.embopress.org/authorguide#revision>). Please insert page numbers in the checklist to indicate where the requested information can be found.
- a letter detailing your responses to the referee comments in Word format (.doc)
- a Microsoft Word file (.doc) of the revised manuscript text
- editable TIFF or EPS-formatted single figure files in high resolution (for main figures and EV figures)

I look forward to seeing a revised version of your manuscript when it is ready. Please let me know if you have questions or comments regarding the revision.

REFeree REPORTS

Referee #1:

In this work, Mortimer and colleagues provide evidence that Ezh2 (within the PRC2 complex) is repurposed in cancer cells downstream of oncogenic signalling to alter key developmental pathways that favor tumorigenesis. By performing ChIP-seq of Ezh2 and H3K27me3 during the stepwise process of untransformed cells to transformed cells, they identify a rather small subset of direct Ezh2 targets that nevertheless have a very strong impact on the ability of cells to generate tumours.

Overall, this is a very interesting work that nicely and elegantly shows that important role of Ezh2 in oncogenic transformation. Although the model to study glioblastoma chosen by the authors might be a bit surprising (i.e. fibroblasts, instead of neural cells) they in fact nicely show its relevance through experiments later done using glioma PDXs and established cells lines (as well as publicly available transcriptomic data). I think the paper will be of interested to the fields of epigenetics and cancer and if minor changes are made I would strongly support its publication in The EMBO Journal.

Main Issues :

- 1) In the introduction the authors describe a sequential mode of action between PRC2 and PRC1. I am aware that this has been, and still is, the prevailing model highly accepted in the field of epigenetics. However, there are interesting new papers showing that this might not always work this way, and that in fact, in some instances, PRC1 is recruited to chromatin before PRC2. I think the authors should at least briefly mention these works to convey to the readership the complexity of these epigenetic mechanisms.
- 2) Although the paper is very well written, I found the section on the potential role(s) of Ezh2 in glioma a bit confusing. On the one hand, the authors say that in glioma patients Ezh2 shows loss-of-function mutations (Brennan et al., 2013), whereas immediately after they say that Ezh2 is required for glioma progression and resistance to therapy (Lee et al., 2008. Suva et al., 2009 etc). I think this apparent discrepancy should be better explained both in the introduction and discussion sections.
- 3) The cell-of-origin of human gliomas has been recently described. Although I think this still maintains the validity of the experimental approach the authors have chosen (that is, to use fibroblasts as a cellular model to obtain information about glioblastoma), the paper in which glioma-initiating cells were described should be mentioned and cited.
- 4) Quantification and statistics should be provided for Figure 1B. Also, the number of biological replicates should be mentioned.
- 5) I am not sure how were the differential peaks called if they only used 2 biological replicates for the ChIP-seq of Ezh2^o and H3^oK27me3. Normally, to obtain statistically relevant results, one should use at least 3 biological replicates and apply DeSeq2 to call differential peaks. The authors should consider adding a third biological replicate and perform this stringent statistical test.
- 6) the results regarding Ezh2 sensitive versus insensitive genes depending on the status of DNA methylation are very interesting and important. The potential interplay between H3K27me3 and DNA demethylation is still quite controversial in the epigenetic field. I think the authors should highlight these results by mentioning them in the abstract and discuss them further.
- 7) 50% of the Ezh2 peaks that are sensitive to Ezh2i locate at promoters. Where do the other 50% locate at? Enhancers, non-coding regions? It would be nice if this data is provided. If they locate at enhancers, would these also regulate neural pathways? Would there be an overlap between enhancer-driven and promoter-driven Ezh2 regulation?
- 8) Statistical significance should be provided for Figures 3F, G, and H
- 9) Quantification and statistics should be provided for Figure 4B.
- 10) It is very difficult to distinguish the precise location within the tumors that are being depicted in Figures 4B and 4E. A counterstain, or H&E stainings side by side should be added to allow the reader to understand which part of the tumors one is looking at. Also, are the HOXB9+ cells

preferentially located at particular areas of the tumours (i.e. next to blood vessels or some immune infiltrate, etc)?

11) It would be desirable if the authors perform orthotopic injections rather than subcutaneous inoculations. This would be a more physiologically relevant experiment regarding tumour growth of the glioblastoma cells with or without the expression of Emx2 (shown in Figure 5).

Referee #2:

This manuscript proposes a very interesting hypothesis that is not well executed. The manuscript has a focus on EZH2, which is well recognized as important in many cancer types, including glioblastoma. The idea of redistribution of EZH2 during transformation is creative and potentially useful, but the authors markedly over-interpret the studies, fail to integrate proper models and controls, and have underpowered studies. While I have enthusiasm for the efforts, the vast majority of the studies would require extensive revision.

Major concerns:

1. The entire consideration of cancer biology is highly simplistic. The authors lack an understanding of transformation generally and brain tumors specifically. Transformation is not a single process. The starting point (cell-of-origin) and mutational and non-genetic events differ markedly between cancer types and even within cancer. The repeated claims that findings are relevant with minimal evidence and experimental effort is noticeable.
2. The authors claim that fibroblasts with sequential transformation as a model for transformation is simply unacceptable. The tumors that result are not gliomas.
3. The authors use a single fibroblast with a single effort to transform these cells. No replicates are used. The stepwise transformation is acceptable, but H-ras mutations do not occur in gliomas. They should use appropriate mutational events. There are many sophisticated genetic models of gliomas. This is not one of them.
4. The claim that neuronal gene marks are affected is validation of the model is incorrect. Long neuronal genes represent a large percentage of the absolute coding sequence. The analysis of EZH2 binding is overly simple.
5. The chromatin studies are not particularly deep and should include more marks.
6. The use of established cell lines to test the role of EZH2 is poorly considered. While they measure expression in a number of established cell lines, the authors reference several papers that show that EZH2 is important in glioblastoma that use patient-derived models. The entire manuscript should be using proper models.
7. The inverse relationship between EZH2 and EMX2 is likely valid but the authors selectively use datasets. Rembrandt is a problematic resource without consideration of the mixed tumor genetics included. The correlation in TCGA is much less modest and EMX2 does not rank near the top of EZH2 correlated genes in any dataset.
8. The biologic effects for the entire manuscript depend on the section entitled, "EZH2-mediated repression of EMX2 is required for glioblastoma maintenance." This claim lacks validity. All that they have done is to show that forced expression of EMX2 reduces proliferation in two cell lines and flank tumor growth in one line. The connection to EZH2 is entirely lacking. They should perform rescue studies, not isolated knockdown. The studies lack clarity in cell biology in molecular mechanism. The *in vivo* tumors are not analyzed, lack replicates, and are in the incorrect environment. The entire figure is minimal.

Minor concerns:

1. Statistical testing and replicates are not well discussed and performed.
2. The referencing throughout the manuscript is biased and many of the papers are not properly interpreted.

Referee #3:

In the manuscript entitled "Rewiring of developmental programs by wild-type EZH2 in cancer cells"

Mortimer and Scaffidi described the key role of EZH2 protein for GBM development and maintenance in both in vitro ('synthetic GBM cell') and in vivo models. EZH2 was shown to be widely distributed throughout the genome of cells after induction of the malignant phenotype, a process that corrupts homeotic genes involved in neural identity. The effect of EZH2 redistribution directly targets EMX2-HOX balance and might be considered an interesting target for future therapies.

For this manuscript, there are several aspects that require clarification and/or precision:

1. For the ChIP experiment in the Fig 3F, although IgG is frequently used as a control, it is not very accurate. It is better to randomly select one or more genomic sequences that are predicted/expected not to bind EZH2 and perform a ChIP with EZH2 antibody (not IgG).
2. The immunofluorescence in Fig 4B shows a very high variation in detection. Very few cells express a lot, others are (nearly) devoid. A better picture (and higher magnification) must be shown.
3. The immunofluorescent images must depict the relevance of HOXB9/EMX2/EZH2 showing their detection in border areas (transition between healthy and cancer tissue).
4. Due to the heterogeneous expression of HOXB9, double staining for EMX2 and HOXB9 should be performed.
5. The difference in EMX2 expression (microarray assay) is not clear enough when comparing to the enormous differences measured in the cell lines and transformation model.
6. The levels of H2K27me3 (fig 2A) seems to be slightly reduced compared with the blot (fig 1B). Qualification and normalization against H3 are missing in this blot (fig2A).
7. The label of figure 3H for the black bars ('mRNA increase') seems incorrect. mRNA levels would be more precise.
8. The correlation between EMX2 and EZH2 ($r=-0.52$) is not as strong as expected. It is borderline significant.
9. The methods do not state clearly how many animals were used.
10. For figure S2F, they state in the text that the pattern was similar among 54 cell lines. However, this does not seem the case for the cell lines above 30 (high variability and overlapping gene methylation levels between EZH2i sensitive and insensitive).
11. The authors explain that heterogeneous HOXB9 expression in FISH analysis accounts for the small difference in the upregulation at the mRNA profile of bulk samples. That seems a wrong assumption since EMX2 shows exactly the same heterogeneous expression by FISH but levels are higher in the mRNA profile.

1st Revision - authors' response

15 July 2019

RESPONSE TO REVIEWERS

Referee #1:

In this work, Mortimer and colleagues provide evidence that Ezh2 (within the PRC2 complex) is repurposed in cancer cells downstream of oncogenic signalling to alter key developmental pathways that favor tumorigenesis. By performing ChIP-seq of Ezh2 and H3K27me3 during the stepwise process of untransformed cells to transformed cells, they identify a rather small subset of direct Ezh2 targets that nevertheless have a very strong impact on the ability of cells to generate tumours.

Overall, this is a very interesting work that nicely and elegantly shows that important role of Ezh2 in oncogenic transformation. Although the model to study glioblastoma chosen by the authors might be a bit surprising (i.e. fibroblasts, instead of neural cells) they in fact nicely show its relevance through experiments later done using glioma PDXs and established cells lines (as well as publicly available transcriptomic data). I think the paper will be of interested to the fields of epigenetics and cancer and if minor changes are made I would strongly support its publication in The EMBO Journal. We thank the reviewer for the positive feedback on our study and for their recommendation to publish our findings.

Main Issues :

1) In the introduction the authors describe a sequential mode of action between PRC2 and PRC1. I am aware that this has been, and still is, the prevailing model highly accepted in the field of epigenetics. However, there are interesting new papers showing that this might not always work this way, and that in fact, in some instances, PRC1 is recruited to chromatin before PRC2. I think the authors should at least briefly mention these works to convey to the readership the complexity of these epigenetic mechanisms.

We agree that the mechanisms underlying Polycomb-mediated repression are complex and we have revised the introduction on p. 3-4 to better reflect this.

2) Although the paper is very well written, I found the section on the potential role(s) of Ezh2 in glioma a bit confusing. On the one hand, the authors say that in glioma patients Ezh2 shows loss-of-function mutations (Brennan et al., 2013), whereas immediately after they say that Ezh2 is required for glioma progression and resistance to therapy (Lee et al., 2008. Suva et al., 2009 etc). I think this apparent discrepancy should be better explained both in the introduction and discussion sections.

We have edited the relevant text and clarified the seemingly contradictory evidence that EZH2 and PRC2 appear to act both as tumour-suppressive and tumour-promoting factors in GBM on p. 4-5. This has also allowed us to reinforce the basis for our hypothesis, namely that EZH2 acts differently in normal (tumour-suppressor) and malignant cells (tumour-promoter)

3) The cell-of-origin of human gliomas has been recently described. Although I think this still maintains the validity of the experimental approach the authors have chosen (that is, to use fibroblasts as a cellular model to obtain information about glioblastoma), the paper in which glioma-initiating cells were described should be mentioned and cited.

We thank the reviewer for raising this. Our findings are in strong agreement with Lee et al., who showed that human glioblastoma originates from astrocyte-like neural stem cells of the periventricular region, where EZH2 and EMX2 are normally co-expressed and ensure controlled self-renewal. We now cite and discuss the recent article on p. 16 as suggested by the reviewer.

4) Quantification and statistics should be provided for Figure 1B. Also, the number of biological replicates should be mentioned.

We have added the requested information in Fig. 1B and the corresponding legend.

5) I am not sure how were the differential peaks called if they only used 2 biological replicates for the ChIP-seq of Ezh2^o and H3^oK27me3. Normally, to obtain statistically relevant results, one should use at least 3 biological replicates and apply DeSeq2 to call differential peaks. The authors should consider adding a third biological replicate and perform this stringent statistical test.

We apologise for the confusion regarding our ChIP-seq analysis. Identification of robust differential peaks, which was not performed using DeSeq2, entailed five steps: i) peak calling on each replicate using SICER; ii) identification of peaks commonly detected in each set of replicates; iii) selection of EZH2 peaks overlapping with H3K27me3 peaks; iv) identification of EZH2 peaks uniquely detected in each cellular state by intersection of peak sets; v) selection of high-magnitude differential peaks based on ChIP-seq tag counts in each cellular state using the 'getDifferentialPeaks' function from Homer (Fig. 1D). Step i) to iv) do not rely on identification of genomic regions showing statistically different EZH2 signal, but on identification of reproducibly detected peaks. In step v) we calculated a p-value for tag enrichment at each peak relative to adjacent genomic regions based on a Poisson distribution. Altogether, the approach we used does not require three replicates. We have amended the **Methods** section to clarify this.

Regardless of the above clarification, we have in fact generated a third EZH2 ChIP-seq replicate to confirm the robustness of the identified differential peaks. We found that 87% of differential peaks were also detected in the third replicate, confirming the validity of our computational approach. We had not included the third replicates in the original manuscript for consistency with H3K27me3, for which we have only two replicates. However, to address the reviewer's comment, we have now added the additional data in the revised manuscript in Fig. EV1E.

6) the results regarding Ezh2 sensitive versus insensitive genes depending on the status of DNA methylation are very interesting and important. The potential interplay between H3K27me3 and DNA demethylation is still quite controversial in the epigenetic field. I think the authors should highlight these results by mentioning them in the abstract and discuss them further.

We agree with the reviewer about the importance of our findings, especially in light of conflicting data reported in the literature. We have added a dedicated paragraph in the **Discussion** on **p. 15** to highlight the results and to attempt to reconcile seemingly contradictory observations made in distinct systems. We have not mentioned the DNA-methylation results in the abstract as we felt it would disrupt the focus on the HOX-EMX2 switch.

7) 50% of the Ezh2 peaks that are sensitive to Ezhi locate at promoters. Where do the other 50% locate at? Enhancers, non-coding regions? It would be nice if this data is provided. If they locate at enhancers, would these also regulate neural pathways? Would there be an overlap between enhancer-driven and promoter-driven Ezh2 regulation?

We thank the reviewer for the suggested analysis. We have compared the differential EZH2 binding sites with active enhancers that we had previously mapped in untransformed and transformed cells by H3K27ac ChIP-seq (excluding promoter regions) (GEO series GSE101242). We find that less than 5% of differential peaks overlap with enhancer regions, with no significant difference detected between untransformed and transformed cells.

Because of the minimal presence of differential EZH2 binding sites at enhancers, we thought the comparison would not provide useful information and we have not included the analysis in the manuscript. However, we have found that a large fraction of the differential peaks overlap with lamina-associated domains, indicating that genomic regions located at the nuclear periphery are particularly affected by transformation-driven changes in H3K27me3 levels. This is particularly interesting in light of studies showing that changes in gene-lamina interactions reflect cell identity (Peric-Hupkes et al. Mol Cell 2010; See et al. Development 2019; Poleshko et al. Cell 2017). We show this new analysis in **Fig S1G**.

8) Statistical significance should be provided for Figures 3F, G, and H
We have added this information.

9) Quantification and statistics should be provided for Figure 4B.
We have added this information in the new **Fig. EV5D**.

10) It is very difficult to distinguish the precise location within the tumors that are being depicted in Figures 4B and 4E. A counterstain, or H&E stainings side by side should be added to allow the reader to understand which part of the tumors one is looking at. Also, are the HOXB9+ cells preferentially located at particular areas of the tumours (i.e. next to blood vessels or some immune infiltrate, etc)?

We have included H&E staining in the new **Fig. 4F** and **Fig. EV5C** indicating the approximate location of the region probed by RNA-FISH in Fig. 4G and Fig. EV5C. As shown in **Fig. EV5C**, we found no clear association between HOXB9+ cells and specific histological features: HOXB9+ cells are found in necrotic areas, next to vessels and in regions with no clear histological features. Conversely, HOXB9+ and HOXB9- cells can be found in regions of similar histology. It is therefore

possible that heterogeneous expression of *HOXB9* may be the result of stochastic processes occurring in the absence of EZH2-mediated repression of the *HOX* clusters. Alternatively, genetic differences among subclones may underlie *HOXB9* expression patterns within tumours.

11) It would be desirable if the authors perform orthotopic injections rather than subcutaneous inoculations. This would be a more physiologically relevant experiment regarding tumour growth of the glioblastoma cells with or without the expression of *Emx2* (shown in Figure 5).

We agree that generation of brain tumours through orthotopic injection of GBM cells is a more physiologically relevant approach to examine the tumour-suppressive role of *EMX2*. We had not been able to perform this experiment because our animal project licence did not permit intracranial injections. We have now amended the licence and performed the suggested experiment, assessing tumour formation by MRI. The results fully confirm our initial findings, showing that *EMX2* expression prevents tumour formation. We show these new results in Fig. 5C-D.

Referee #2:

This manuscript proposes a very interesting hypothesis that is not well executed. The manuscript has a focus on EZH2, which is well recognized as important in many cancer types, including glioblastoma. The idea of redistribution of EZH2 during transformation is creative and potentially useful, but the authors markedly over-interpret the studies, fail to integrate proper models and controls, and have underpowered studies. While I have enthusiasm for the efforts, the vast majority of the studies would require extensive revision.

Major concerns:

1. The entire consideration of cancer biology is highly simplistic. The authors lack an understanding of transformation generally and brain tumors specifically. Transformation is not a single process. The starting point (cell-of-origin) and mutational and non-genetic events differ markedly between cancer types and even within cancer. The repeated claims that findings are relevant with minimal evidence and experimental effort is noticeable.

We agree with the reviewer that cancer mechanisms are diverse and it is impossible to recapitulate the complexity of the disease in one experimental model, especially because any model has, by definition, limitations. Our approach was to start from a model system that, while reductionist in nature, allowed direct comparison of non-tumorigenic and tumorigenic human cells, and subsequently examine a large number of cancer cell lines and patient datasets to find evidence that the identified mechanism is active in the human disease. We do find general support for our experimental findings in patient and patient-derived cell lines, even in the absence of any stratification based on likely cell-of-origin or genetic drivers. This suggests that the observed changes, while not universal, can be broadly observed. We are unclear what the reviewer would consider sufficient evidence that our findings are relevant. We have now added additional explanation on p. 6 to clarify these points.

2. The authors claim that fibroblasts with sequential transformation as a model for transformation is simply unacceptable. The tumors that result are not gliomas.

As stated above, the fibroblast-based model is only a discovery tool. While imperfect, it uncovered an EZH2-mediated switch in homeotic gene expression that we have subsequently confirmed by analysing normal and malignant neural cells and GBM patients. While GBM murine models exist, we wanted to characterize EZH2 behaviour in human adult cells. Obtaining human normal neural cells from adult individuals is extremely challenging. Astrocytes commercially available from Lonza are of foetal origin and are not suitable, as paediatric and adult GBM have extensive differences in their clinical, genetic and epigenetic features, especially with regards to H3K27me3 (Jones et al. Neuro-oncology 2017). Furthermore, most available datasets and cell lines are from adult patients. Deriving neural cells from human iPSCs and subsequently transforming them may have provided a more physiologically relevant system, but would have introduced significant technical hurdles, which may have compromised the overall success of the study. In addition, this approach would have not solved the problem of uncertainty regarding the disease cell-of-origin. Using the fibroblast-based system was therefore a compromise that allowed us to formulate hypotheses using a tractable system, which we then tested in more physiologically relevant contexts. We have clarified these points at p.6.

We note that referee 1 acknowledges the relevance of the model system, based on the confirmations obtained from GBM cells and patients, and referee 3 does not question its use.

3. The authors use a single fibroblast with a single effort to transform these cells. No replicates are used.

We believe that analysis of normal and cancerous neural cells and human patients is a more stringent validation of the initial findings than performing experimental transformation with other samples. To address the issue of different drivers, we have assessed whether the EZH2 -EMX2 relationship correlates with any particular genetic background in GBM patients and found similar patterns regardless of the molecular subtype, indicating that multiple drivers can induce EZH2 redistribution and aberrant *EMX2* silencing. We show the result of this analysis in **Fig. EV4B** (see also the point below and point 7).

The stepwise transformation is acceptable, but H-ras mutations do not occur in gliomas. They should use appropriate mutational events. There are many sophisticated genetic models of gliomas. This is not one of them.

While mutations in RAS genes are indeed not frequent in GBM, activation of the RAS pathway is highly common, as a result of aberrant activation of upstream components such as EGFR or PDGFR or inactivation of the RAS regulator NF1 (Verhaak, et al. Cancer Cell 2010). We clarify this point on **p. 6**. As mentioned above, we have chosen to use a reductionist approach at the start our study, and direct activation of RAS allowed us to recapitulate downstream effects of EGFR, PDGFR and NF1 mutations.

4. The claim that neuronal gene marks are affected is validation of the model is incorrect. Long neuronal genes represent a large percentage of the absolute coding sequence.

We apologise for the confusion. This analysis is limited to peaks overlapping gene transcription start sites, hence, gene length is not relevant. Furthermore, since the GO analysis accounts for differential representation of a given gene class in the genome, the enrichment is significant. We have clarified this point in the **Methods** section.

The analysis of EZH2 binding is overly simple.

We are unsure what the referee is referring to and what type of analysis is requested.

5. The chromatin studies are not particularly deep and should include more marks.

The study focuses on EZH2 and we analyse the relevant histone mark, H3K27me3, which is deposited by EZH2. We are unsure what other histone marks would increase the significance of our findings.

To address **point 7 of referee 1**, we have compared differential EZH2 binding sites with active enhancers mapped by ChIP-seq of H3K27ac, but found only minimal presence of differential peaks at enhancers.

6. The use of established cell lines to test the role of EZH2 is poorly considered. While they measure expression in a number of established cell lines, the authors reference several papers that show that EZH2 is important in glioblastoma that use patient-derived models. The entire manuscript should be using proper models.

While we agree that patient-derived xenografts are excellent model systems, we also note that extensive literature uses established glioblastoma cell lines combined with patient data to uncover mechanisms of glioma development (Masamha et al. Nature 2014; Kim D et al. Nature 2015; Mancini et al. Cancer Cell 2018, as a few examples).

To reinforce the relevance of established lines with respects to our findings, we show that the pattern of gene misregulation across the four *HOX* clusters is highly similar in cell lines and patients (**Fig. 4B** and **Fig. EV3B**). Since *HOX* genes are established targets of EZH2, this will serve as an indirect assessment of EZH2 localisation and activity.

7. The inverse relationship between EZH2 and EMX2 is likely valid but the authors selectively use datasets. Rembrandt is a problematic resource without consideration of the mixed tumor genetics included. The correlation in TCGA is much less modest and EMX2 does not rank near the top of EZH2 correlated genes in any dataset.

We primarily used the REMBRANDT dataset as it contains the largest number of normal controls, which are essential for assessing *HOX* genes and *EMX2* misregulation in glioma. The glioma TCGA dataset confirms a significant inverse correlation between *EZH2* and *EMX2* ($p < 0.001$), as seen in the REMBRANDT and Chinese Glioblastoma Atlas data sets. It also shows a significant correlation of *EMX2* levels with grade ($p < 0.001$). In addition, the inverse correlation between *EZH2* and *EMX2* is retained in tumours classified by their transcriptional profiles (mesenchymal, proneural, classical and neural), which are highly dependent on the genetic drivers (Verhaak, et al. Cancer Cell 2010). We show this data in **Fig. EV4 A,B,D**.

We do not believe that *EMX2* failing to rank at the top of *EZH2* correlating genes weakens our conclusions. It must be noted that *EZH2* levels are affected by the proliferative index of tumours and this could introduce confounding effects when assessing correlations with other genes (Bracken et al. EMBO J 2003, Wassef et al. Gene & Dev 2015).

8. The biologic effects for the entire manuscript depend on the section entitled, "EZH2-mediated repression of *EMX2* is required for glioblastoma maintenance." This claim lacks validity. All that they have done is to show that forced expression of *EMX2* reduces proliferation in two cell lines and flank tumor growth in one line. The connection to *EZH2* is entirely lacking. They should perform rescue studies, not isolated knockdown.

We agree the subheading overstated our results and we have amended it accordingly.

The experiment shown in Fig. 5B aims at testing the importance of *EMX2* silencing in DBTRG-05MG cells, by mimicking the effect of its de-repression. The connection with *EZH2* lies in our observation that *EMX2* silencing in those cells is entirely dependent on *EZH2* activity (Fig. 3G). Rescue experiments are unlikely to succeed as *EMX2* is only one of several genes repressed by *EZH2* and it is unlikely that knock-down of only one gene will rescue the overall effect of *EZH2* inhibition. Our results show that *EMX2* silencing in DBTRG-05MG cells is *necessary* to preserve tumorigenic potential. Even if it is not *sufficient* on its own, we believe the significance of our finding stands. We will clarify these points in the revised manuscript on **p. 13**.

The studies lack clarity in cell biology in molecular mechanism.

We are unclear what the reviewer refers to

The in vivo tumors are not analyzed,

Tumours do not grow when *EMX2* is expressed. We are thus unsure what type of analysis the reviewer expects.

lack replicates,

As stated in the **figure legend**, the values shown in Fig. 5B are averages of 6 injections per condition, and we have repeated the experiment twice, using biological replicates (two transgenic lines infected independently at different times).

and are in the incorrect environment. The entire figure is minimal.

We agree that generation of brain tumours through orthotopic injection of GBM cells is a more physiologically relevant approach to examine the tumour-suppressive role of *EMX2*. We had not been able to perform this experiment because our animal project licence did not permit intracranial injections. We have now amended the licence and performed the suggested experiment, assessing tumour formation by MRI. The results fully confirm our initial findings, showing that *EMX2* expression prevents tumour formation. We show these new results in **Fig. 5C-D**.

Minor concerns:

1. Statistical testing and replicates are not well discussed and performed.

We had stated the statistical test used and the number of replicates in all figure legends and in Methods. We had omitted to indicate a p-value in a few cases where differences were of great magnitude and significance was apparent. We have now added this in every panel.

2. The referencing throughout the manuscript is biased and many of the papers are not properly interpreted.

We have carefully reviewed the references that we cite and added a few additional ones. It would have been helpful if the reviewer had indicated the papers that we may have improperly interpreted.

Referee #3:

Comments for the authors

In the manuscript entitled "Rewiring of developmental programs by wild-type EZH2 in cancer cells" Mortimer and Scaffidi described the key role of EZH2 protein for GBM development and maintenance in both in vitro ('synthetic GBM cell') and in vivo models. EZH2 was shown to be widely distributed throughout the genome of cells after induction of the malignant phenotype, a process that corrupts homeotic genes involved in neural identity. The effect of EZH2 redistribution directly targets *EMX2*-*HOX* balance and might be considered an interesting target for future therapies.

For this manuscript, there are several aspects that require clarification and/or precision:

♣ For the ChIP experiment in the Fig 3F, although IgG is frequently used as a control, it is not very accurate. It is better to randomly select one or more genomic sequences that are predicted/expected not to bind EZH2 and perform a ChIP with EZH2 antibody (not IgG)

Figure 3F does contain such a control: *GAPDH*, which, as expected, is not bound by EZH2.

♣ The immunofluorescence in Fig 4B shows a very high variation in detection. Very few cells express a lot, others are (nearly) devoid. A better picture (and higher magnification) must be shown. We agree the image originally used was not ideal due to the low magnification. That was chosen to better show the heterogeneous expression of *HOXB9*, but we agree the signal was difficult to see. We now provide higher magnification images and a quantification of the staining in the new Fig. EV5 C,D.

We also added additional data regarding *HOX* genes in patients in Fig. 4B, Fig. EV3 C,D. While *HOXB9* gene expression changes were particularly striking in the transformation model, a general de-repression of *HOX* genes is observed in patients, with other *HOX* genes showing in fact higher upregulation. We have now revised the figures to better emphasise this aspect.

♣ The immunofluorescent images must depict the relevance of *HOXB9*/*EMX2*/*EZH2* showing their detection in border areas (transition between healthy and cancer tissue)

Unfortunately, the samples that we have analysed do not contain a clear tumour-normal border.

While we are unable to perform the requested FISH analysis, we note that Fig. 4D addressed the referee's point with regards to *EMX2* and *EZH2*, showing that micro-dissected areas of tumour and normal adjacent tissue probed by RNA-seq show opposite expression patterns. We have now extended the analysis to *HOX* genes in Fig. EV3C and found upregulation of numerous genes in tumour cells compared to the normal adjacent tissue.

♣ Due to the heterogeneous expression of *HOXB9*, double staining for *EMX2* and *HOXB9* should be performed.

We now show double staining for *EMX2* and *HOXB9* and corresponding quantification in Fig. EV5 C,D. We show that expression of *HOXB9* does not correlate with histological features, suggesting its heterogeneous expression may be due to either stochastic processes occurring in the absence of EZH2-mediated repression, or subclonal genetic differences within individual tumours. We also show that regardless of *HOXB9* expression, *EMX2* levels are consistently low across tumours.

♣ The difference in *EMX2* expression (microarray assay) is not clear enough when comparing to the enormous differences measured in the cell lines and transformation model.

The milder repression of *EMX2* in the REMBRANDT dataset has multiple possible explanations:

i) Tumours may contain contaminating normal tissue that expresses *EMX2*.

ii) Data from the transformation model and cell lines are obtained through RNA-seq while the REMBRANDT dataset is based on microarrays, which are characterized by background noise that limits detection of truly silenced genes. In agreement, RNA-seq-based data from the Chinese Glioma Atlas show a more robust repression, especially in grade 4 patients.

ii) Since EZH2 is upregulated in proliferating cells (Bracken. et al., EMBO J 2003, Wassef et al. Gene & Dev 2015), its overall levels, and consequently *EMX2* repression, depend on how many cells within a tumour are actively proliferating. Thus, a low proliferative index may result in apparently incomplete repression at the bulk level. We explain these points in the **legend of Fig. 4A**.

♣ The levels of H2K27me3 (fig 2A) seems to be slightly reduced compared with the blot (fig 1B). Qualification and normalization against H3 are missing in this blot (fig2A). The slight discrepancy between the two western blots was likely due to small difference in loading, masked by saturation of the H3 signal in Fig. 2A. We have repeated the Western blot and now provide quantification using unsaturated images in **Fig. 2A**.

♣ The label of figure 3H for the black bars ('mRNA increase') seems incorrect. mRNA levels would be more precise. We apologise for the confusion. What is plotted in the graph is the fold change in mRNA levels upon EZH2i treatment relative to the DMSO control, not actual levels. We now explain this in the **figure legend**.

♣ The correlation between *EMX2* and EZH2 ($r=-0.52$) is not as strong as expected. It is borderline significant. We are unsure why the referee considers the correlation as borderline significant. The p-value of the Spearman rank correlation is highly significant ($p < 0.0001$). It must be noted that variable presence of non-cancerous cells (normal adjacent tissue and infiltrates) across samples may influence the observed correlation.

♣ The methods do not state clearly how many animals were used. The number of replicates used for *in vivo* assays, and the number of independent experiments performed was indicated in the figure legend. We have now also added the information in the **Methods**.

♣ For figure S2F, they state in the text that the pattern was similar among 54 cell lines. However, this does not seem the case for the cell lines above 30 (high variability and overlapping gene methylation levels between EZH2i sensitive and insensitive). We agree that our statement was imprecise and we have corrected this. We have also indicated statistical significance in the figure.

♣ The authors explain that heterogeneous *HOXB9* expression in FISH analysis accounts for the small difference in the upregulation at the mRNA profile of bulk samples. That seems a wrong assumption since *EMX2* shows exactly the same heterogenous expression by FISH but levels are higher in the mRNA profile. We apologise for the confusion. The expression pattern of *EMX2* and *HOXB9* in GBM sections is very different: *HOXB9* is expressed in patches across the tumour, while *EMX2* is broadly repressed, and only expressed in very few, isolated cells. We now show this more clearly in **Fig. EV5 C,D**. This expression pattern is in agreement with the overall reduction in *EMX2* levels observed in microarray datasets (lower median value, $p < 0.0001$) and overall increase in *HOXB9* (higher median value, $p < 0.01$). If the referee refers to the few patients that show high *EMX2* levels in Fig. 4A, those are clearly outliers and not representative of the whole cohort.

Regardless of the point discussed above, as outlined earlier, we have modified **Fig. 4B** and **Fig. EV3 B-D** to more appropriately show that the entire *HOX* cluster, not just *HOXB9*, is de-repressed in GBM.

2nd Editorial Decision

29 July 2019

Thank you for the submission of your revised manuscript to our editorial offices. We have now received the reports from the two referees that were asked to re-evaluate your study, you will find below. As you will see, both referees support the publication of your study in EMBO reports. Referee #2 has some final remarks and suggestions, I ask you to address with an appropriate text

changes in a final revised version of your manuscript.

Further, I have a few editorial requests:

- The title presently reads rather like one from a review article. Could you provide a more active and comprehensive title? E.g.:
Wild-type EZH2 rewires developmental programs to promote cancer.
- Please name the items in the Appendix as in the TOC (i.e. Appendix Figure Sx, Appendix Table Sx ...).
- Please remove the referee tokens from the final version of the paper, and assure that the deposited data will be public upon publication of the paper.
- Finally, please find attached a word file of the manuscript text (provided by our publisher) with changes we ask you to include in your final manuscript text, and some queries, we ask you to address. Please provide your final manuscript file with track changes, in order that we can see the modifications done.

In addition I would need from you:

- a short, two-sentence summary of the manuscript
- two to three bullet points highlighting the key findings of your study
- a schematic summary figure (in jpeg or tiff format with the exact width of 550 pixels and a height of not more than 400 pixels) that can be used as a visual synopsis on our website.

REFEREE REPORTS

Referee #1:

In this revised version of the manuscript the authors have addressed all the main issues I raised. I think it is of particular importance that they have performed the orthotopic models that I suggested which have greatly strengthened the *in vivo* relevance of their conclusions. Considering this I now strongly recommend this work for publication in EMBO reports. I think it will be of high interest to the broad readership of this journal, specially those specializing in epigenetics and cancer research.

Referee #2:

The manuscript has been improved and rebuttal explains the changes made.

I remain very sceptic about the fibroblast experiments. Their justification is reasonable but not entirely convincing. It takes a large part of the first paragraph of the result section which in my opinion would have been better placed in the intro in which a lot of information is redundant.

I am somewhat critical in the way these fibroblast and other cell line experiments are eluded to in the text. The cells are not clearly specified and it does not get entirely clear which cells are referred to.

Finally, and notwithstanding my skepticism with respect to how they are uncovered, the observation of the inverse correlation with EMX and its general appearance in many cancer cell lines is interesting and justifies publication.

To address Referee2's remaining concerns we have removed redundant sentences in the introduction and more clearly specified the identity of the cell lines used in each experiment/analysis. As agreed by email, we have kept the paragraph justifying the choice of the model system in the results section, as we believe it makes it easier for readers to understand the rationale behind our approach.

Corresponding Author Name: Paola Scaffidi

Journal Submitted to: EMBO reports

Manuscript Number: 2019-48155